# Unveiling the Nutritional Profile and Safety of Coffee Pulp as a First Step in Its Valorization Strategy

**DOI:** 10.3390/foods13183006

**Published:** 2024-09-22

**Authors:** Alicia Gil-Ramírez, Miguel Rebollo-Hernanz, Silvia Cañas, Ignacio Monedero Cobeta, Pilar Rodríguez-Rodríguez, Andrea Gila-Díaz, Vanesa Benítez, Silvia M. Arribas, Yolanda Aguilera, María A. Martín-Cabrejas

**Affiliations:** 1Department of Agricultural Chemistry and Food Science, Faculty of Science, Universidad Autónoma de Madrid, C/Francisco Tomás y Valiente, 7, 28049 Madrid, Spain; alicia.gil@uam.es (A.G.-R.); miguel.rebollo@uam.es (M.R.-H.); silvia.cannas@uam.es (S.C.); vanesa.benitez@uam.es (V.B.); yolanda.aguilera@uam.es (Y.A.); 2Institute of Food Science Research (CIAL, UAM-CSIC), C/Nicolás Cabrera, 9, 28049 Madrid, Spain; 3Food, Oxidative Stress and Cardiovascular Health (FOSCH) Research Group, Universidad Autónoma de Madrid, 28049 Madrid, Spain; ignacio.monedero@uam.es (I.M.C.); pilar.rodriguezr@uam.es (P.R.-R.); andrea.gila@uam.es (A.G.-D.); silvia.arribas@uam.es (S.M.A.); 4Department of Physiology, Faculty of Medicine, Universidad Autónoma de Madrid, C/Arbobispo Morcillo, 2, 28029 Madrid, Spain

**Keywords:** coffee pulp, food ingredient, dietary fiber, amino acids, minerals, bioactive compounds, caffeine, (poly)phenols, safety, toxicity

## Abstract

The coffee pulp, a significant by-product of coffee processing, is often discarded but has potential for recycling and high-value uses. This study aimed to investigate the chemical composition of two coffee pulp ingredients, a flour (CPF) and an aqueous extract (CPE), and conducted acute and sub-chronic toxicity assays to determine their safety. The proximate composition revealed the high fiber content of both ingredients; the CPF mainly contained insoluble fiber, while CPE consisted exclusively of soluble pectic polysaccharides. The CPF had higher concentrations of amino acids and a better balance of essential/non-essential amino acids, whereas the CPE exhibited higher concentrations of free amino acids, ensuring higher bioavailability. Both ingredients showed elevated mineral content, while heavy-metal concentrations remained within acceptable limits. This study established the bioactive potential of the CPF and the CPE, demonstrating the high content of caffeine and gallic, protocatechuic, and 4-caffeoylquinic acids. The toxicity studies revealed that the CPF and the CPE exhibited safety when orally administered to mice. Administered doses were non-toxic, as they did not induce lethality or adverse effects in the mice or produce significant histopathological or biochemical adverse changes. This study represents a first step in valorizing the CPF and the CPE as safe novel food ingredients with health benefits for functional and nutritional foods.

## 1. Introduction

The global coffee industry is one of the most significant agricultural sectors, with an annual production of millions of tones [1]. The large-scale production of coffee beans simultaneously generates a considerable quantity of by-products, among which coffee pulp is a significant constituent [2]. The coffee pulp, comprising the outermost layer of the coffee cherry and representing approximately 41% of the cherry’s weight, is produced during the wet processing when the pulp and skin are removed after washing [3]. The coffee pulp has traditionally been regarded as a waste product, with around 1 ton generated for every 2 tons of green coffee beans produced [4]. However, this by-product is increasingly recognized for its potential contribution to valorization and sustainability in the food industry, making its effective utilization a matter of economic and environmental interest [5]. Existing proximate chemical studies reveal that coffee pulp is rich in dietary fiber, protein, and minerals [6]. Nevertheless, the comprehensive nutritional profile of the coffee pulp, particularly its amino acid profile and mineral composition, remains largely unexplored [7]. Furthermore, this by-product contains an array of phenolic compounds and other compounds, such as caffeine, widely known as bioactive compounds promoting improved health statuses [8]. Recent studies have shown that these phytochemicals are bioaccessible and potentially bioavailable, showing stability and absorption during in vitro digestive processes [9]. The coffee pulp has also reported promising inhibitory effects on adipogenesis, obesity-related inflammation, mitochondrial dysfunction, and insulin resistance. Additionally, it can regulate human metabolism, modulating mitochondrial bioenergetics and lipid and glucose metabolism [10,11]. Altogether, these effects open exciting opportunities for using coffee pulp as a bioactive food ingredient or nutraceutical, offering benefits in preventing and managing chronic metabolic disorders.

The potential of this coffee by-product as a bioactive food ingredient is not only conditioned on its nutritional value but also its safety for consumption [12]. The European Food Safety Authority (EFSA) is involved in assessing coffee by-products’ safety and regulatory status in the context of their classification as novel foods within the European Union. The EFSA has confirmed the safety of the coffee pulp, implementing regulations specifying the use of the dried coffee pulp for preparing infusions as traditional foods [13]. Nevertheless, the commercial trading of coffee pulp and its processed derivatives in the European market still depends on further novel food approvals and additional risk assessments [14]. Given that the regulatory approval of this by-product as a novel food is currently under examination, it is imperative to conduct comprehensive scientific risk assessments to establish its safety. In addition, the contaminants present and their bioavailability in coffee pulp are necessary to assess potential risks to human health. Their bioavailability and occurrence, however, are influenced by other factors such as soil pH, cation-exchange capacity, organic matter content, soil texture, and interactions among elements [15]. These evaluations should be completed by performing extensive toxicological studies, encompassing both acute and sub-chronic toxicity analyses, which are essential to determine its feasibility as a food ingredient [4].

The effective valorization of the coffee pulp would represent a considerable advance toward achieving sustainability in the coffee industry. The shift from considering coffee pulp as a waste to a valuable resource could trigger innovation in creating new, nutrient-rich, and safe food products [16]. For this upcycling strategy to materialize, validating the coffee pulp’s nutritional profile and safety is essential as a first step in a stepwise approach to valorizing it as a bioactive food ingredient or nutraceutical [17]. While the potential of the coffee pulp as a bioactive ingredient has been recognized, specific aspects, such as its amino acid and mineral profiles and toxicological safety, have not been previously examined in two key ingredients from the coffee pulp: flour (CPF) and extract (CPE). Therefore, this study aimed to fill those gaps in the knowledge of coffee pulp by validating its nutritional value and assessing its safety for consumption, thereby establishing a foundation for its potential valorization as a bioactive food ingredient. The comprehensive characterization of its chemical composition and acute and sub-chronic toxicity assays will provide a more holistic understanding of the coffee pulp’s potential for application in the food industry.

## 2. Materials and Methods

### 2.1. Sample Preparation

The coffee pulp used in this study was sourced from the “Las Morenitas” farm, located in the highlands of Nicaragua, northwest of Jinotega, at an altitude of 1200 m (13.2082–85.8871). The pulp was mechanically separated from the cherries of the *Arabica* species, the variety Caturra, through the wet processing method. After collection, the raw and sun-dried coffee pulp was packaged in sealed bags and shipped at room temperature. The coffee pulp was ground into flour in a pilot-scale ball mill (Ortoalresa-Álvarez Redondo S.A., Madrid, Spain) for 72 h to produce the CPF and stored in sealed containers at −20 °C until needed. The CPE was derived from the CPF using a previously optimized extraction process [18]. Shortly, the flour (0.02 g mL^−1^ solid-to-solvent ratio) was added to boiling water at 100 °C and stirred continuously for 90 min. Afterward, the solution was filtered, frozen at −20 °C, and freeze-dried. The resultant CPE was preserved in sealed containers at −20 °C until further use. From a single batch of coffee pulp (25 kg), 5 kg was milled to produce the CPF, which was subsequently used to prepare the CPE. All replications for the analysis were performed using this single batch of coffee pulp.

### 2.2. Nutritional and Chemical Composition

Proximate chemical composition analysis, including crude protein, fat, and total ash, was assessed following official AOAC procedures [19]. Total carbohydrates were calculated by difference, and available carbohydrates were determined by subtracting the dietary fiber from the total carbohydrates. The energy content was estimated using the Atwater factors (2 kcal g^−1^ for fiber, 4 kcal g^−1^ for protein and available carbohydrates, and 9 kcal g^−1^ for fat). Total dietary fiber (TDF), the sum of insoluble dietary fiber (IDF) and soluble dietary fiber (SDF) was determined by the enzymatic-gravimetric method (Mes-Tris AOAC method 991.43) with slight modifications [20]. The polysaccharide sugar composition was analyzed after acid hydrolysis, subjecting the insoluble fiber residues to 12 mol L^−1^ H_2_SO_4_ treatment for 3 h at room temperature followed by dilution to 0.6 mol L ^−1^ H_2_SO_4_ hydrolysis at 100 °C for 3 h (Seaman hydrolysis). HPLC-PAD was employed to analyze the neutral sugar composition in CPF and CPE. Hydrolysates were neutralized using AG4-X4 resin, and sugars were analyzed using a microguard column (Aminex Carbo-P) in series with a carbohydrate analysis column (Aminex HPX-87P). Sugars were quantified using standard sugars, with erythritol as the internal standard [20]. The concentration of uronic acids was determined using a commercial kit (K-URONIC, Megazyme Co., Wicklow, Ireland). 

### 2.3. Amino Acid Composition 

#### 2.3.1. Amino Acid Extraction

For the extraction of total amino acids, CPF (~40 mg) and CPE (~15 mg) were mixed with 200 µL of 6 mol L^−1^ HCl and subjected to 110 °C for 21 h to achieve acid hydrolysis of the proteins [21]. Afterward, the samples were vacuum-dried, weighed, and stored at −20 °C for further analysis. Norleucine (6 nmol for CPF and 20 nmol for CPE) was added as an internal standard before hydrolysis to normalize amino acid recovery.

The extraction of free amino acids was performed as previously reported, with minor adjustments [22]. Briefly, 150 mg of CPF or CPE were frozen with N_2_ in a mortar. Immediately after, the samples were homogenized in the presence of 600 μL of a solution of H_2_O:chloroform:methanol (3:5:12, *v*/*v*) while maintaining the frozen state of the vegetable material. The mixture was centrifuged at 12,000 rpm for 2 min. The supernatant was collected, and the residue was re-submitted to the same solid–liquid extraction process. Then, both supernatants were mixed with 300 μL of chloroform and 450 μL of ultrapure water, vigorously shaken for 3 min, and centrifuged at 12,000 rpm for 2 min. The hydroalcoholic phase was collected and dried by a gentle N_2_ stream. Samples were kept at −20 °C. 

#### 2.3.2. Amino Acid Analysis

The instrumental analysis was carried out entirely by the external service. A specific amount of the total amino acid extracts was resuspended in the loading buffer (6% for CPF and 20% for CPE). For free amino acid analysis, samples were resuspended in a 500 µL loading buffer. Thus, the samples (extracts of total and free amino acids) were examined by ion exchange chromatography and post-column online derivatization with ninhydrin using a Biochrom 30+ amino acid analyzer following manufacturer instructions (Biochrom, Cambridge, UK). Amino acids were identified based on the retention times of their corresponding standards, and their quantification was performed by integrating their absorbance signals at 570 nm (440 nm for proline), prepared with known amounts of amino acid standards submitted to acid hydrolysis. Note that for total amino acids, the data for glutamic acid corresponded to the glutamine and glutamic acid sum; likewise, the aspartic acid corresponds to the sum of asparagine and aspartic acid. Tryptophan was not determined due to its sensitivity to acid hydrolysis, which causes degradation of its indole ring.

#### 2.3.3. Estimation of Protein Quality 

The essential amino acid index (EAAI) and the amino acid score (AAS) were selected to assess the protein quality of CPF and CPE and calculated using the following equations:(1)$$\mathrm{EAAI}\left(\%\right)=n^{\log\mathrm{EAA}}$$
(2)$$\log\mathrm{EAA}=\left[\frac{1}{n}\right]\times \left[\log\left(\frac{100a_{1}}{a_{1R}}\right)+\log\left(\frac{100a_{2}}{a_{2R}}\right)\ldots +\log\left(\frac{100a_{n}}{a_{\mathrm{nR}}}\right)\;\right]$$
where n corresponds to the number of amino acids contemplated for the estimate (n = 8), a denotes the mg of the amino acid g^−1^ of protein and a_R_ indicates the mg of the amino acid g^−1^ of reference protein. Higher EAAI values indicate better protein quality, with values below 75% considered inadequate [23].
(3)$$\mathrm{ASS}\;\left(\%\right)=100\;\times \;\frac{\frac{\mathrm{mg}\;\mathrm{AA}}{g\;\mathrm{protein}\;\mathrm{tested}}}{\frac{\mathrm{mg}\;\mathrm{AA}}{g\;\mathrm{of}\;\mathrm{reference}\;\mathrm{protein}}}$$
where the value of the denominator corresponds to those defined by FAO, WHO and UNU [24], 15, 30, 59, 45, 16, 6, 22, 38, 23, and 39 mg AA g^−1^ of protein for histidine, isoleucine, leucine, lysine, methionine, cysteine, methionine + cysteine, phenylalanine + tyrosine, threonine, and valine, respectively. Results around 100% stands for a quality protein as high as the standard is for the specific amino acid. The ASS of protein corresponded to the lowest ASS value within the essential amino acids.

The sum of total conditionally essential amino acids (TCEAs), the branched-chain amino acids (BCAAs), the total aromatic amino acids (TarAAs), the total sulfur amino acids (TSAAs), the total basic amino acids (TBAAs), and total acid amino acids (TAAAs) were calculated. In addition, the sum of antioxidants (histidine, lysine, methionine, tyrosine, and proline), immunomodulatory (histidine, arginine, and glutamic acid), and cytotoxic (cystine, histidine, aspartic acid, and proline) amino acids were determined. The Fischer ratio, calculated as the ratio of BCAA to TarAA, and the Lys/Arg ratio were estimated.

The predicted protein efficiency ratios (PERs) are calculated using the following equations:p-PER1 = −0.684 + 0.456 × Leu − 0.047 × Pro(4)
p-PER2 = −0.468 + 0.454 × Leu − 0.105 × Tyr(5)
p-PER3 = −1.816 + 0.435 × Met + 0.78 × Leu + 0.211 × His − 0.944 × Tyr(6)

The predicted Nutritional Index (p-NI) and the Biological Value (p-BV) were calculated to provide additional insights into the protein quality. The equations used were:p-NI = EAAI × protein content percentage(7)
p-BV = 1.09 × EAAI − 11.7(8)

### 2.4. Elemental Composition

#### 2.4.1. Digestion Treatment

For the elemental determination in CPF and CPE matrices, 100 mg of dried powder were placed into porcelain crucibles in duplicate. Samples were introduced in a muffle furnace (J.P. Selecta, Barcelona, Spain) and submitted to 180, 280, and 380 °C for 45 min each, followed by 2 h at 480 °C before cooling in a desiccator overnight. Afterward, ashes were carefully mixed with ultrapure water (2.5 mL) and an 18.5% HCl solution, covered with a glass watch to avoid material losses, and submitted to 80 °C for 30 min. Finally, the acid-digested samples were collected with ultrapure water and filtered using a Whatman grade 41 quantitative ashless filter paper. Samples were diluted with ultrapure water up to 10 mL final volume for further elemental analysis. Moreover, several digestions without CPF or CPE material were carried out for their consideration as sample blanks. Ashes yield was calculated by gravimetry. The procedure was performed in triplicate. 

#### 2.4.2. Inductively Coupled Plasma Optical Emission Spectrometry (ICP-OES) Analysis

The elemental analysis was performed by Inductively Coupled Plasma Optical Emission Spectroscopy (ICP-OES iCAP PRO, Thermo Fisher Scientific, Waltham, MA, USA) equipped with an auto-sampler model ASX-560. Data acquisition and processing were conducted using the Thermo Scientific™ Qtegra™ Intelligent Scientific Data Solution™ (ISDS) software 2.7. The plasma was vertically oriented and dually viewed except for Ca, Mg, and K, which were detected by radial view. Pure Ar 99.99% (1 and 2.4 L min^−1^ for dual and radial view, respectively) and a radio-frequency power of 1500 W were used for plasma generation. Likewise, Ar was also used as an auxiliary and sample nebulizer. Samples (0.5 mL min^−1^) were simultaneously scanned for determination of four macroelements (Ca, Mg, P, and K), six microelements (B, Cu, Fe, Mn, Si, and Zn), and five contaminants (As, Cd, Cr, Pb, and Ni). The quantitative analysis was accomplished at the following spectral lines: Ca, 315.887 nm; Mg, 279.806 nm; P, 177.495 nm; K, 769.896 nm; B, 208.959 nm; Cu, 324.754 nm; Fe, 259.940 nm; Mn, 257.610 nm; Si, 251.611 nm; Zn, 202.548 nm; As, 189.042 nm; Cd, 228.802 nm; Cr, 267.716 nm; Pb, 220.353 nm; and Ni, 221.647 nm.

Calibration curves were performed using individual standards for As, Si, and P diluted in a 2% HNO_3_ aqueous solution and a multi-elemental standard solution for the other mentioned elements under study. The ranges of the calibration curves corresponded to 0–5 µg mL^−1^ for Si, P, K, Mg, and Ca; 0–10 µg mL^−1^ for As; and 0–2.5 µg mL^−1^ for B, Cu, Fe, Mn, Zn, Cd, Cr, Pb and Ni. The analyses for both standard solutions and samples were performed in triplicate. A 30 s washing was settled between two successive samples to eliminate memory effects. The content of the mentioned elements was measured in triplicate for both digested samples and blanks.

### 2.5. Phytochemical Profile Analysis

#### 2.5.1. Extraction

Samples were prepared as previously described [25]. Briefly, CPF and CPE (1.0 g) were mixed with 50 mL of methanol-HCl (1‰)/water (80:20, *v*/*v*), submitted to ultrasounds for 30 min, and finally shacked for 16 h at 40 °C. The supernatants were collected after centrifugation (4000× *g*, 4 °C, 15 min). This process was performed twice prior to evaporation to dryness under vacuum. Samples were solubilized in 10 mL methanol and kept at −20 °C until analysis.

#### 2.5.2. HPLC-DAD-ESI/MSn Analysis of Phenolic Compounds and Methylxanthines 

For (poly)phenols and methylxanthines analysis, a Hewlett-Packard-1100 HPLC chromatograph equipped with a diode array detector (DAD) and a quaternary pump, made by Agilent Technologies in Palo Alto, CA, USA, was utilized [9]. The mobile phases were 0.1% formic acid in water (solvent A) and 100% acetonitrile (solvent B). The elution gradient was set as follows: isocratic 15% B for 5 min, 15–20% B for 5 min, 20–25% B for 10 min, 25–35% B for 10 min, 35–50% B for 10 min, followed by column re-equilibration. Chromatographic separation of phytochemicals was carried out using a Spherisorb S3 ODS-2 C8 column (Waters, Milford, MA, USA) measuring 3 µm, 150 mm × 4.6 mm i.d., at a flow rate of 0.5 mL min^−1^ and a temperature of 35 °C. The DAD detection was performed at 280 nm for hydroxybenzoic and hydroxycinnamic acids and caffeine and 370 nm for flavonols and flavones. The mass spectrometer (MS), coupled to the HPLC apparatus via the DAD cell output, carried out detection using an API-3200 Qtrap (Applied Biosystems, Darmstadt, Germany), which is outfitted with an ESI source, a triple quadrupole-ion trap mass analyzer, and Analyst 5.1 software. The identification of phytochemicals was achieved by combining their retention times, UV and mass spectra, fragmentation patterns, and comparison with authentic standards when available. Additionally, for acyl-quinic acids, the positions of substitutions were determined by employing the recommended IUPAC numbering system and using hierarchical keys. In the quantitative analysis, the quinic derivatives of acyl acids were quantified using calibration curves of the corresponding free acids; for C-glycoside flavones derived from apigenin, apigenin 6-C-glucoside (isovitexin) was utilized as a reference. The quantification of quercetin derivatives was performed using the calibration curve of quercetin-3-*O*-glucoside. For methylxanthines, its quantification was based on its standard calibration curves. Each phytochemical’s concentration was expressed in milligrams per 100 g (mg 100 g^−1^) of the sample. 

### 2.6. Toxicity Assays in Experimental Animals

Eight-week-old adult mice (C57B1/6 J) from a colony maintained at the animal house facility of Universidad Autónoma de Madrid (ES-28079-0000097) were used. Mice breeding and experimental procedures conformed to the Guidelines for the Care and Use of Laboratory Animals (National Institutes of Health publication no. 85-23, revised in 1996), the Spanish legislation (RD 53/2013), and the Directive 2010/63/EU on the protection of animals, were approved by the Ethics Review Board of the Universidad Autónoma de Madrid and by the Regional Environment Committee of the Comunidad Autónoma de Madrid (PROEX 108/19, Madrid, Spain). Mice were grouped by sex in groups of 5 animals. Environmental parameters were controlled in the animal holding room with a temperature of 22 ± 1 °C a humidity of 40 ± 4%, and the light/dark period was set at 12:12 h. Mice were fed ad libitum with a standard diet containing 51.7% carbohydrates, 21.4% protein, 5.1% lipids, 3.9% fiber, 5.7% minerals, and 12.2% humidity (SafeA03; Safe Augy, France). The diet was the same for all groups, with only the supplementation differing: the control group received plain gelatin cubes, while the CPF and CPE groups were administered gelatin cubes containing CPF or CPE. Fresh drinking water was available ad libitum. 

#### 2.6.1. Preparation of the CPF and CPE Supplementation Animal Assay

Gelatin cubes, prepared with 100% bovine gelatin, were chosen as CPF and CPE administration vehicles [26]. Gelatins containing CPF or CPE were employed for the acute and sub-chronic toxicity assays (2000 and 1000 mg kg^−1^, respectively). A voluntary ingestion training procedure was needed before mice supplementation. The animals were trained with neutral gelatin cubes until full acceptance by placing the animal in an empty box without bedding for 2 h. This procedure was carried out for 5 days per week at the same hour. The first day that the mice ingested the entire gelatin cube was considered the acceptance day.

#### 2.6.2. Acute Toxicity and Sub-Chronic Toxicity Tests

For the acute toxicity supplementation assay, the diet of 5 trained males and 5 females was supplemented with a single dose of 2000 mg kg^−1^ CPF (OECD No. 425). Likewise, neutral gelatins were administrated for supplementation controls (*n* = 5 by gender group). The mice were kept under observation for 14 days. During this period, changes in body weight and behavior were monitored by OECD guidelines, including changes in skin and fur, eyes and mucous membranes, and respiratory, circulatory, autonomic, and central nervous systems, as well as somatomotor activity and behavior patterns. On day 14, the mice were weighed and then humanely euthanized with CO_2_ exposure, followed by exsanguination. The liver, kidney, intestine-duodenum, spleen, heart, and thymus were carefully harvested from each mouse. The tissues (liver, kidney, and intestine) were then subjected to a detailed histopathological analysis. Initially, the tissues were fixed in a 10% formalin solution to preserve the cellular architecture. The tissues were embedded in paraffin wax blocks following fixation to facilitate sectioning. Thread sections (approximately 5 µm thick) were cut from the paraffin-embedded tissues using a microtome. These sections were then mounted on glass slides and stained with hematoxylin and eosin. Hematoxylin stains the cell nuclei blue, while eosin stains the cytoplasm and extracellular matrix pink, providing a clear contrast that aids in examining tissue structure and identifying pathological changes under a light microscope.

The sub-chronic supplementation protocol was carried out in adult female mice (*n* = 5) for 90 days. The daily dose was 1000 mg kg^−1^ of CPF and CPE. Five females were supplemented with neutral gelatin cubes as controls (OECD No. 408). During this period, body weight and behavior changes were monitored by OECD guidelines. On day 90, the mice were weighed and then humanely euthanized with CO_2_ exposure followed by exsanguination. Afterward, the previously described organs were collected and weighted, and tissue histopathological analyses were performed. Blood was centrifuged for 10 min (900× *g* at 4 °C), and the plasma was aliquoted and stored at −80 °C. To confirm that supplementation unaffected liver and kidney function, we analyzed blood samples externally at Laboratorio Animales Exóticos 24 h (Madrid, Spain). Biochemical parameters, including liver function markers (alkaline phosphatase (ALP), alanine aminotransferase (ALT), aspartate aminotransferase (AST), gamma-glutamyl transferase (GGT), and total bilirubin) and renal function tests (albumin, total protein, creatinine, blood urea nitrogen, inorganic phosphorus, and calcium), were measured using a portable biochemical analyzer (VETSCAN VS2^®^, Zoetis, Parsippany, NJ, USA). Assessment of total, low-density lipoprotein cholesterol (LDL) and high-density lipoprotein (HDL) cholesterol, and the quantification of insulin and leptin were determined using specific kits following the manufacturer’s protocols (Spinreact, Barcelona, Spain; BioVendor R&D R, Brno, Czech Republic). Assessment of total, low-density lipoprotein cholesterol (LDL) and high-density lipoprotein (HDL) cholesterol, and the quantification of insulin and leptin were determined using specific kits following the manufacturer’s protocols (Spinreact, Barcelona, Spain; BioVendor R&D R, Brno, Czech Republic).

### 2.7. Statistical Analysis

Data were analyzed by one-way analysis of variance (ANOVA) and post-hoc Tukey HSD test for comparisons within the same CPF or CPE matrix. *T*-test comparisons were performed between CPF and CPE. To compare against the control group, which consisted of mice non-supplemented in toxicity assays, a one-way (ANOVA) and post-hoc Dunnett’s test were conducted. Differences were considered significant at *p* < 0.05. The graphs and statistical analyses were conducted using GraphPad Prism 8.0 (San Diego, CA, USA).

## 3. Results

### 3.1. Proximate Analysis Showed High-Fiber Content and Nutritional Potential of CPF and CPE

The coffee pulp extract (CPE) showed a high extraction yield (51.7%) derived from the coffee pulp flour (CPF) under a previously optimized extraction process. This value is the highest compared to other coffee pulp aqueous extractions (29–14%) [27] as well as other coffee by-products extracts using boiled water such as coffee silverskin (14%) and parchment (2.3%) [12] or other extraction solvents [28]. As described in Table 1, the CPF was mainly composed of carbohydrates (76.7%), followed by protein (12.9%) and fat (2.8%), while CPE showed lower protein content (7.5%) and higher fat (3.8%) but similar carbohydrates (70.9%). The carbohydrate fraction showed lower concentrations of available carbohydrates at 26.9% for CPF and 42.9% for CPE. These available carbohydrates were mainly constituted by free sugars, with fructose being the predominant sugar, comprising approximately 70% of the free sugars in both matrices. CPF showed the highest content of TDF at 49.8% dry weight (dw), with the majority being in the insoluble fraction (38.6%, representing 77.5% of TDF), while the SDF accounted for 11.2%. The SDF:IDF ratio was close to 1:3, consistent with previous studies on coffee pulp [27]. These data suggested that CPF is a suitable source of dietary fiber to be included as a food ingredient, offering the physiological benefits of soluble and insoluble fiber fractions. These benefits include helping to regulate blood sugar levels and lower cholesterol, making CPF a valuable addition to fiber-enriched food products [29]. 

In contrast, CPE, an aqueous extract from CPF, only exhibited SDF (28.0%). Overall, the fiber contents of CPF and CPE are comparable to those of other fruit and vegetable by-products (peels, seeds, and pomace) (38–62%) with excellent physicochemical properties [30]. Indeed, their high fiber content permits considering both matrices as a source of fiber since their content goes over 3 g per 100 g or more [31]. In addition, the main monosaccharide components of CPF fiber polysaccharides were uronic acids (40%), glucose (30%), and arabinose (14%), whereas xylose, galactose, and mannose occurred in lower content (4–6%). Then, pectic polysaccharides, mainly homogalacturonans and arabinans, were inferred to be the primary cell wall polysaccharides, followed by cellulose. In contrast, hemicelluloses (xyloglucans and arabinogalactans) hardly appeared. The CPE acidic and neutral sugar composition showed the predominance of uronic acids (87%) and, to a lesser extent, arabinose (11%), indicating the entire presence of soluble pectic polysaccharides. Thus, these coffee pulp samples can be excellent candidates for prebiotic ingredients being metabolized by probiotic bacteria (*Lactobacillus* and *Bifidobacterium*), producing short-chain fatty acids among other beneficial metabolites [32]. Based on the results, CPF and CPE exhibited a high potential to be included in fermented products, beverages, and fiber-rich functional foods with improved health outcomes, as well as in the bakery and confectionery industry, enhancing the techno-functional properties of these products (swelling and water holding capacity) [33]. Therefore, the SDF:IDF ratio and polysaccharide composition of CPF and CPE revealed a better valorization of coffee pulp, ameliorating their nutritional functionality, palatability, and technological applicability. 

Concerning proteins, a significantly higher amount was recorded for CPF (12.9%) compared to CPE (7.5%), which is in agreement with the literature (7.9–10.7%, depending on the processing method) [34,35]. Not all the CPF proteins were released during the CPE obtention. The diversity of proteins, which possess different physicochemical properties, can influence their solubility. Furthermore, the aqueous extraction can also lead to the denaturation of some proteins, affecting the extraction efficiency [36]. Regarding fats, CPF contained up to 2.8%, while CPE exhibited 3.8%, probably due to the concentration phenomenon, unlike other aqueous coffee pulp extracts prepared under milder conditions [27]. In the case of ash, results from CPF were in line with some authors statements (7.6%), although significantly higher values were observed for CPE (14.6%) (*p* < 0.05) (Table 1), which is intimately related to the macroelements content obtained through the applied hydrothermal extraction process to CPF. However, the energy content was similar, with CPE having higher energy (304.6 kcal) than CPF (284.0 kcal), reflecting its higher fat and protein content.

### 3.2. CPF Exhibited a Superior Amino Acid Profile and Protein Quality Compared to CPE 

The study of the amino acid profiles for CPF and CPE results is relevant to predicting the potential of both agroindustry by-products as a source of quality proteins. Regarding the total amino acid analysis, targeted amino acids were identified in both CPF and CPE; however, different profiles of specific matrices were observed (Table 2). Therefore, methionine was the least abundant amino acid in CPF (15.4 mg 100 g^−1^), while aspartic acid showed the highest concentration (333.2 mg 100 g^−1^) (*p* < 0.05). Likewise, the aspartic acid content was statistically superior in CPE with no differences compared to CPF (333.2 mg 100 g^−1^); however, histidine, isoleucine, leucine, lysine, methionine, cysteine, and tyrosine exhibited the lowest content (*p* < 0.05). The scarce references found in the literature regarding the amino acid content in the coffee pulp point to similar features in their qualitative profiles [37], as is the case for other coffee agricultural by-products such as silverskin [38]. However, the contents obtained in this work were previously reported, which can be justified by the original matrix composition (meaning coffee variety, growing conditions, etc.), besides the extraction and analytical methodologies [37,39]. 

No bibliography was found relative to the total amino acid content in aqueous coffee pulp extracts. However, compared to CPF, lower contents of amino acids were found for CPE, except for arginine, aspartic acid, and cysteine. The differences were not statistically significant between matrices, and proline showed an opposite trend (*p* < 0.05). Several phenomena could explain the differences in the total amino acid profiles of CPF and CPE, such as the uncomplete extraction of protein content from the CPF by the performed heat-assisted extraction derived from the protein denaturation and subsequent water insolubilization, as well as the protein and amino acid degradation at high temperatures previously observed, even for coffee silverskin by-product [40,41]. This observation agrees with the free amino acid content observed for CPE, since aggressive treatments promote their release. The sum of total amino acids (∑AA) observed for CPF (2123.7 mg 100 g^−1^) was significantly higher than that obtained for CPE (1229.3 mg 100 mg^−1^). Likewise, the total essential amino acid (∑EAA) and total non-essential amino acid (∑NEAA) concentrations were superior for CPF (*p* < 0.05). The contribution of the EAA to the total amino acid content (% EAA) for CPF (29.7%) was similar and slightly lower to those published for coffee silverskin and coffee bean powder, respectively [38,42]. Moreover, just 14% of the TAA belongs to EAA in CPE, significantly lower than that observed for CPF (*p* < 0.05). Thus, CPF is more promising than CPE, considering the EAA–NEAA balance.

The interest in analyzing free amino acids in both matrices lies in the absorption kinetics reported for conjugated and free amino acids (understood as availability), resulting in easy amino acid recognition by cellular receptors [43]. Therefore, studying the free amino acid profile for CPF and CPE will notably contribute to roughing the potential of both matrices as protein sources. The content of each targeted free amino acid was significantly lower compared with their respective total amino acid concentrations (*p* < 0.05) for CPF and CPE (Table 2). Proline was the free amino acid, showing the highest (*p* < 0.05) content in both CPF (26.4 mg 100 g^−1^) and CPE (74.2 mg 100 g^−1^). Lysine and methionine were the free amino acids found in the lowest concentrations. Contrary to what was observed for total amino acids, CPE showed a higher content of each free amino acid than CPF, except for methionine (*p* < 0.05). Then, total amino acids (∑AA) were 171.7 mg 100 g^−1^ for CPE and 57.3 mg 100 g^−1^ for CPF. Nevertheless, the contribution of EAA (%EAA) in CPE followed a similar trend to that observed for CPF. The EAA and NEAA distributions in total and free amino acids revealed distinctive EAA/NEAA ratios. The ratio in CPF total amino acids (0.4) was significantly higher than that obtained for CPE (0.2) (*p* < 0.05). The CPF EAA/NEAA ratio resulted lower than that reported for coffee pulp and even for other agricultural coffee by-products, such as silverskin, which can be explained by the composition of the original material itself (i.e., coffee variety, cultivation conditions, etc.) [38,39]. However, less promising results were reached when studying the EAA/NEAA ratio of free amino acids in both matrices (0.2 and 0.1 for CPF and CPE, respectively). 

The essential amino acid index (EAAI), the amino acid score (AAS), amino acid functional groups, ratios, and protein nutritional indices were studied to provide comprehensive insights into the nutritional quality of CPF and CPE based on the amino acids profile (Figure 1). EAAI shows the protein quality by comparing the value of the essential amino acids in the tested sample to a reference protein. AAS approaches the protein quality based on the potential capacity of a food matrix to provide a proper profile of dietary essential amino acids. The amino acid pattern considered as a reference is detailed by FAO/WHO/UNU [24]. In consonance with the amino acid distribution, the EAAI for CPF was higher than that observed for CPE (76.4 and 38.0%, respectively) and like that previously reported for coffee pulp [39] (Figure 1A). Therefore, as agreed by the scientific community, the protein content of CPF showed a positive quality, with EAAIs between 75 and 86%, while the CPE’s protein was labeled as inadequate protein [23]. 

In addition, the pattern of the essential amino acids for CPF expressed as mg of amino acid g^−1^ of protein (Figure 1A) showed that lysine and the sulfur-containing amino acid methionine did not reach the required concentration defined by the reference protein, with ASS of 58 and 45%, respectively (Figure 1B). Regarding CPE, a higher number of amino acids showed a poor ASS, such as the basic amino acids, histidine (53%) and lysine (15%), as well as for the non-polar neutral amino acids, leucine (33%), isoleucine (36%), and methionine (20%). On the contrary, the sum of phenylalanine and its hydroxylation product tyrosine for CPF and cysteine for CPE stood out, reaching double and triple the reference value, respectively. The limiting amino acid, understood as the amino acid showing the highest concentration difference related to the same amino acid in the reference protein, corresponded to methionine for CPF (45.4%) and lysine for CPE (15.0%), meaning that both amino acids covered less than a half of the required concentrations. Considering that the ASS of protein corresponded to the lowest ASS value within essential amino acids, the nutritional value of the proteins in CPF and CPE were those related to methionine and lysine. The calculated ASS of protein for CPF is consistent with those of low-quality coffee green beans and coffee silverskin, as well as of other plant-based proteins [44]. As depicted in Figure 1C, CPF revealed a significantly higher concentration of BCAA, TarAA, TBAA, and the sum of Ile + Leu + Phe + Val compared to CPE, indicating that CPF could better support muscle metabolism, stress-related physiological needs, and overall protein synthesis [45]. Contrarywise, CPE protein comprises a higher level of TCEA, TAAA, and cytotoxic effect amino acids, featuring its potential for supporting condition-specific nutritional needs and cognitive function and providing protective effects against abnormal cell growth [42]. The high content of antioxidant amino acids in CPF indicated its capacity to reduce oxidative stress, whereas the higher concentration of immunomodulatory amino acids in CPE pointed to its immune-system-modulating properties. The Fischer and Lys/Arg ratios, higher in CPF, implied its better potential to regulate metabolism and liver function, improve cholesterol metabolism, and benefit the cardiovascular system (Figure 1D) [43]. Predicted protein efficiency ratios (p-PER1, p-PER2, p-PER3), also significantly higher in CPF, denoted its better protein quality and potential effectiveness in promoting growth and nitrogen balance. These ratios confirm that CPF is a more efficient protein source for dietary requirements. [44]. Ultimately, CPF showed higher predicted nutritional indices (p-NI) and biological values (p-BV) than CPE (Figure 1E), which indicated better overall protein quality and the efficient utilization of CPF’s protein by the body, respectively. Consequently, these indices confirm that CPF is a more nutritionally valuable and bioavailable protein source than CPE. The exhaustive amino acid study revealed that CPF outperforms in terms of amino acid content, functional properties, and overall protein quality, making it an ideal alternative for increasing dietary protein consumption and promoting various positive health effects.

Then, the superior aspartic acid content in TAA must be highlighted since its presence is crucial for both the endocrine and central nervous systems. Overall, CPF contains higher concentrations of amino acids, a more promising balance of EAA/NEAA, and an increased EAAI compared to CPE. However, CPE showed higher concentrations of FAA, facilitating their ready availability but with lower loads of EAA compared to CPF. CPF demonstrated higher levels of BCAA, TarAA, TBAA, and the sum of Ile + Leu + Phe + Val, suggesting better support for muscle metabolism and protein synthesis. Conversely, CPE’s higher content of TCEA, TAAA, and cytotoxic effect amino acids highlights its potential for cognitive function and immune support. The superior Fischer and Lys/Arg ratios, p-PER, p-NI, and p-BV in CPF confirm its overall higher protein quality and bioavailability. These results reinforce the idea of recognizing the protein concentration in CPF as an extra reason for using coffee pulp as a food ingredient due to its association with other compounds of interest from a nutritional and physiological perspective. 

### 3.3. CPF and CPE Revealed a Rich Mineral Composition and a Safety Profile Adverse Minerals 

The coffee pulp is considered a mineral-enriched coffee by-product, standing out over the coffee husk, parchment, and silverskin, with an ash content ranging between 7.5 and 20.3%, according to several authors [34,35]. These notable differences were evidenced by the coffee variety, the chemical composition of the growing soil, and the drying process applied to the coffee pulp after coffee cherry depulping [34,46]. In concordance with the literature, the potassium content in the CPF and the CPE showed a significantly high contribution to the ashes’ weight compared to the other studied macroelements, such as calcium, magnesium, and phosphorous, which were statistically grouped (Table 3). Bridging the differences due to the already mentioned environmental and technological factors, the obtained potassium content for the CPF (24.2 mg g^−1^) was in line with reported data [47], although notably higher than that published by Patil et al., 2022. Moreover, higher calcium and magnesium concentrations in CPF were observed compared to the literature [35]. Regarding the CPE, the potassium concentration was significantly higher (35.1 mg g^−1^) than that found in the CPF. This mentioned superior concentration also occurred with the other analyzed macroelements, excluding the phosphorous for which a half concentration was obtained in the CPE (*p* < 0.05), which denotes a concentration phenomenon derived from the heat-assisted aqueous extraction process applied to the CPF. The adverse health effects of potassium at high concentrations must also be considered. In this regard, the daily supplementation of 103 or 71 g of CPF and CPE on a standard diet background appears to be safe for the healthy population, corresponding to 2500 mg of potassium per day [48]. Likewise, the CPE could also be valuable because of its magnesium content, while lower values were observed for the CPF (Table 3). 

On the contrary, phosphorus content in CPF resulted twice as high as compared with that obtained for CPE (*p* < 0.05). Following these results, one gram of CPE would cover 1.4% of the potassium-adequate intakes for adult women and 0.5% of the recommended dietary allowances for calcium and magnesium of the same population group [48]. However, one gram of CPF would make a more significant contribution to ensuring the recommended dietary allowances of phosphorus compared to CPE [48]. Regarding the elements needed by the body in minimal amounts, silicon was the most abundant microelement, followed by the remaining studied elements (*p* < 0.05) (Table 3). Scarce data related to the coffee pulp’s microelements content can be found in the literature. However, the obtained results for CPF, although higher, were in concordance with those reported for coffee pulp’s iron and zinc concentrations (39.3 and 7.6 µg g^−1^, respectively) [35]. Overall, the observed concentrations were significantly higher in CPF than CPE, except for zinc, indicating that an uncomplete recovery for most of the minerals occurred during the heat-assisted extraction (Table 3). Thus, following the Health Professional Fact Sheets of microelements from the National Institutes of Health [49], lower contributions to the recommended dietary allowances of iron and copper per ingested gram of CPE are expected compared to those observed for CPF (0.5 and 1.9%, respectively). Likewise, the CPF contribution (1 g) to the adequate intake of manganese would be 3.4%, while 0.8% for CPE.

Moreover, the presence of five highly toxic heavy metals in very small amounts, including arsenic, cadmium, chromium, lead, and nickel, was studied in CPF and CPE (Table 3). Arsenic, a cancer-causing inorganic agent, was exclusively detected in CPF (18.2 µg 100 g^−1^) with a concentration slightly higher than reported values for coffee beans and coffee silverskin [50]. Nowadays, no specific legislation defines the allowed content of the studied heavy metals, including arsenic, in coffee and coffee by-products. However, according to Annex 1 in the EU regulation, the maximum levels for certain contaminants in food are described [51]. The obtained data fall below the values claimed for rice flour (25 µg 100 g^−1^). Moreover, following the guidance of CONTAM Panel (the EFSA Panel on Contaminants in the Food Chain) related to the benchmark dose lower confidence limit where the change in response is likely to be smaller than 5% (BMDL_05_), which corresponded to 0.06 µg kg^−1^ bw per day, the CPF’s amount to overcome such limit would result in an exclusive intake up to 23.3 g per day for an averaged European adult body weight of 70.8 kg [52].

Cadmium, a heavy metal capable of causing metabolic abnormalities, bone degradation, or kidney stone formation, was identified in CPF and CPE at 2.5 and 6.1 µg 100 g^−1^, respectively. Likewise, with arsenic, neither coffee beans nor coffee by-products are included in the EU regulation 2023/915, so no reference is available for a trustworthy comparison [51]. However, considering that the tolerable weekly intake (TWI) of cadmium was established in 2.5 μg kg^−1^ bw (being 24.8 μg per day for a 70.8 kg averaged European adult body weight), the exclusive maximum daily intake of CPF and CPE for an adult would result up to 992 g and 406 g, respectively [53].

Chromium is a needed element that plays a relevant role in the macromolecule’s metabolism. However, its overconsumption can lead to hypoglycemia, dyslipidemia, and organ injury, which is relevant to controlling its diet levels and elucidating the contribution of potential new ingredients to its total daily intake [54]. CPF showed a similar chromium content to CPE, the most abundant element of the targeted heavy metals for both matrices (Table 3). Chromium levels in CPF and CPE aligned with previous reports on coffee beans [55]. To exceed chromium’s tolerable daily intake (TDI), an exclusive intake of 15.5 or 23.0 kg per day of CPF and CPE, respectively, for a 70.8 kg average European adult body weight should happen [56].

Lead is considered a contaminant connected to misfunctioning of the central nervous system and was not detected in CPF but in CPE. Nevertheless, CPE’s content was shallow (0.5 μg 100 g^−1^) (Table 3). These results denoted lower lead contents in the matrices under interest than those reported for green coffee beans and coffee silverskin [57]. Nor coffee nor coffee by-products are included as food sources in the EU regulation 2023/915 on the maximum levels for certain contaminants in food (Annex 1). However, the lead concentration in CPE is below that established for all the food sources included in the norm [51]. According to the lead level in CPE, the exclusive maximum intake would result in around 7.0 kg per day.

Nickel, which is pointed out as an allergenic, carcinogenic, and organ-misfunctioning contaminant, was also present in CPE and CPF (Table 3). The published information on the nickel content in green coffee beans shows a high variability. The nickel concentrations obtained for CPF and CPE were in harmony with the reported data, even considering the levels in coffee silverskin [55]. According to EFSA’s scientific opinion about the total amount of nickel that can be taken daily over a lifetime without appreciable health risk (TDI = 13 μg kg^−1^ bw per day), the amounts of 873 g per day and 1.7 kg per day of CPF and CPE, respectively, should be exclusively consumed to exceed such mentioned value [58].

Therefore, the mineral composition supports the strategy of using CPF and CPE as potential food ingredients because of their remarkable content of macro- and microelements compared to other usually consumed food matrices. Regarding the elements needed in high amounts (over 20 mg per day), the CPE showed more promising results with a total of 42.3 mg g^−1^ compared to CPF (28.7 mg g^−1^), covering up the 1.4% of the potassium-adequate intakes for adult women by the consumption of 1 g of CPE. Overall, the coffee pulp, or its derived matrices, showed low levels of contaminants, so similar potential toxicity could be assumed for CPF and CPE compared to the foodstuff mentioned. Most contaminants were present in such low concentrations that very abundant food intakes must occur to reach their corresponding TDI and TWI values. However, the arsenic concentration must be considered since it appears to be the limiting contaminant that can be used to establish a daily consumption of CPF. 

### 3.4. CPE Exhibited a Higher Concentration of (Poly)phenols and Caffeine than CPF

The phenolic compounds and methylxanthines identified in coffee pulp through HPLC-DAD-MS^n^ analysis are presented in Table 4. The analysis in coffee pulp led to the identification of 14 compounds grouped into five distinct groups. Among these, the hydroxybenzoic acid group comprised two compounds, hydroxycinnamic acid derivatives encompassed seven compounds, flavones were represented by one compound, flavonols by three compounds, and methylxanthines by one compound. The concentration of total phenolic compounds reached 356.4 mg 100 g^−1^ in CPF and 597.6 mg 100 g^−1^ in CPE. Concerning CPF, hydroxybenzoic acids (222.6 mg 100 g^−1^) constituted the most abundant group of phenolic compounds, with protocatechuic acid being the predominant component observed, exhibiting a concentration of 175.7 mg 100 g^−1^, followed by gallic acid (46.9 mg 100 g^−1^). These compounds comprised more than 62.5% of the determined (poly)phenols. Hydroxycinnamic acids represented the second major group in terms of concentration (110.8 mg 100 g^−1^), as they presented the most remarkable diversity of compounds, with a total of 7. Within this group, 4-caffeoylquinic acid, *trans*, was the most prevalent compound with a concentration of 85.3 mg 100 g^−1^. Several studies have identified chlorogenic acid (5-caffeoylquinic acid) as the major hydroxycinnamic acid in coffee pulp, followed by 4-caffeoylquinic, and 3-caffeoylquinic acids [34,59]. This variance in the concentration of chlorogenic acid and other bioactive compounds may be influenced by several factors, including coffee species or variety, the degree of ripeness, local climate, soil characteristics, agricultural practices, and post-harvest processing methods [60]. The coffee pulp used in this study was sourced from the “Las Morenitas” farm in the highlands of Nicaragua, situated at an altitude of 1200 m with a tropical humid climate and temperatures ranging from 15 to 26 °C year round. These environmental factors and sustainable agricultural practices significantly impact the coffee pulp quality and biochemical composition. Soil conditions, such as nutrient availability, organic matter, and pH, along with the surrounding vegetation, such as cedar and guava trees, contribute to the microenvironment that affects the chemical composition of the coffee pulp, influencing its flavor and nutritional profile through subtle metabolic interactions [61,62].

Apigenin-6,8-di-C-glucoside was the only flavone found, presenting a 6.4 mg 100 g^−1^ concentration. Among the flavonols, quercetin-3-*O*-rutinoside was identified as the most abundant, with a concentration of 7.9 mg 100 g^−1^. This was closely followed by quercetin-3-*O*-glucoside (6.1 mg 100 g^−1^), although there was no significant difference in their levels. Quercetin-3,7-di-*O*-glucoside was also detected in a smaller amount. Among the flavonoids analyzed, catechin and epicatechin were not detected. However, these flavanols are expected to be found in coffee pulp, particularly epicatechin, which usually exhibits a high concentration [63,64]. 

Lastly, the methylxanthine detected in CPF was caffeine, found at a high concentration (473.1 mg 100 g^−1^). Similarly to chlorogenic acid, the caffeine content in coffee and its by-products may vary depending on the coffee’s origin, extraction and analysis methods, degree of roasting, coffee processing, and environmental factors, such as climate and altitude where the coffee is grown, among others [34]. Regarding CPE, the hydroxybenzoic acids also constituted the predominant group with a concentration of 380.7 µg 100 mg^−1^. Regarding CPE, the hydroxybenzoic acids also constituted the predominant group with a concentration of 380.7 mg 100 g^−1^. In this group, protocatechuic acid was the primary compound, exhibiting the highest concentration with 312.2 mg 100 g^−1^. Similarly, gallic acid was also found at 68.5 mg 100 g^−1^. Five hydroxybenzoic acids were identified in the CPE, compared to seven found in CPF. Among the detected hydroxybenzoic acids, 4-caffeoylquinic acid, *trans*, exhibited a concentration of 145.0 mg 100 g^−1^, thereby being the predominant compound, like in CPF. Other compounds, such as 3-caffeoylquinic, 4-caffeoylquinic, 5-feruloylquinic, and 3,5-dicaffeoylquinic acids, were identified in lower quantities. 5-Caffeoylquinic acid, reported in the literature as the main hydroxycinnamic acid in coffee pulp, was not detected in CPE [27]. Apigenin-6,8-di-C-glucoside was the only flavone detected, reaching a concentration of 8.0 mg 100 g^−1^. Regarding flavonols, quercetin-3-*O*-rutinoside (14.5 mg 100 g^−1^) and quercetin-3-*O*-glucoside (8.9 mg 100 g^−1^) were reported. Like in CPF, caffeine was the primary compound identified in CPE, achieving an elevated concentration of 787.9 mg 100 g^−1^. Caffeine was the only alkaloid detected in CPF and CPE in this work. However, other alkaloids have been identified in other coffee pulp studies, including theobromine and trigonelline [27]. The total phenolic compounds and caffeine concentrations were significantly higher in CPE than in CPF. The high-temperature aqueous extraction effectively released the phenolic compounds from the CPF matrix [18]. Food processes, such as hot water extraction, help to break down the cell wall matrix by depolymerizing pectin and hemicelluloses, thereby facilitating the release of these compounds bound to the fibers of CPF and allowing them to dissolve in water [65]. On the other hand, the difference between the number of phenolic compounds found in CPF (13) and those found in CPE (10) could be due to the protective effect of the food matrix. Despite the high temperatures during the aqueous extraction, some compounds may remain intimately bound to the fiber matrix and not be released into the extract [66]. Non-extractable phenolic compounds include high-molecular-weight proanthocyanins, hydrolyzable tannins, flavonoids, and low-molecular-weight phenolics. They can be found cross-linked to components such as cellulose, pectin, hemicelluloses, lignin, and structural proteins by covalent bonds [65]. Moreover, the extraction of phenolic compounds from the CPF matrix may have been influenced by the solubility of these compounds. Since water was the solvent used, it is important to note that (poly)phenols tend to be more soluble in less polar organic solvents than in water, as reported in the literature [67]. Overall, the diversity and distribution of bioactive compounds in coffee pulp are consistent with findings in other coffee by-products, such as coffee silverskin, parchment, husk, and spent coffee grounds. Studies have highlighted caffeine and chlorogenic acids as predominant components in these by-products. For instance, coffee silverskin and husk aqueous extracts contain caffeine levels of 1922 and 982 mg/100 g and chlorogenic acid concentrations of 279 and 346 mg/100 g, respectively [68]. Acetone–water extracts from spent coffee grounds report caffeine and chlorogenic acid levels of 83 and 147 mg/100 g [69], which are comparable to the concentrations observed in our study. Additionally, coffee husk demonstrated a rich phenolic profile, including significant amounts of protocatechuic and gallic acids [68]. In contrast, coffee parchment exhibited a lower chlorogenic acid content (29 mg/100 g) but contained other notable phenolic compounds, such as vanillic and *p*-coumaric acids [70]. These results highlight the consistent presence of phenolic compounds and caffeine across coffee by-products, reinforcing the potential of coffee pulp, particularly CPF and CPE, as valuable ingredients for functional foods and nutraceuticals. The higher phenolic and caffeine concentrations in CPE suggest its suitability for developing value-added products, especially for their antioxidant and stimulant properties.

### 3.5. CPF and CPE Showed No Toxic Effects in Acute and Sub-Chronic Toxicity Studies

Food toxicity studies are often used to assess potential risks to human health from naturally occurring plant chemical constituents that may cause adverse effects. Acute toxicity assessments may provide limited clinical data, whereas sub-chronic toxicity tests offer sufficient data for food and drug development and in vivo and clinical studies. Therefore, this study is the first to evaluate the acute and sub-chronic toxicity of CPF and CPE in C57BL/6 J mice, which serves as the basis for subsequent experiments by this research group. 

During the 14 days of observation after the acute administration of CPF and CPE (2000 mg kg^−1^), no behavioral alterations indicative of distress or mortality were observed in any male or female mice. Furthermore, there were no evident abnormal physical or clinical signs, and no mortality was observed. In male mice of both supplemented groups, weight and weight gain at day 14 (CPF = +0.8 g; CPE = +0.5 g) was similar to control mice (+1.2 g) (Figure 2A,B). In female mice, weight gain was observed on day 14 in both supplemented groups (CPF = +1.0 g; CPE = +1.2 g), which was comparable to the control group (+0.8 g) (Figure 2C,D). Thus, male and female mice supplemented with CPF and CPE and non-supplemented control mice gained weight comparable to average values. Body weight is determined by many factors, including the state of experimental animals, growth hormone levels, food intake, neurotransmitters, and environmental factors [71]. In male mice, livers of CPF- and CPE-supplemented mice were significantly smaller than in control mice. No significant differences were detected in the weight of the remaining organs (Figure 2E). In female mice supplemented with CPF or CPE, no significant differences were detected in the relative weights of heart, liver, kidney, and spleen compared to control mice (Figure 2F). Relative organ weight is one of the most sensitive parameters about chemically induced changes in organs and is considered a routine assessment to clarify any toxic effects on organs, especially the liver [72]. Although organ weight changes are an important marker in toxicological research because they can suggest potential toxicity or deleterious consequences, they must be interpreted in conjunction with other findings, such as histopathology. Hence, histopathological liver, kidney, and intestine analyses were performed (Figure 2G). The liver was normal in male and female mice, and the kidney showed no significant abnormalities. Liver histology in control and treated groups displayed normal hepatic lobules, polyhedral hepatocytes with central vesicular nuclei, and eosinophilic granular cytoplasm. No overt signs of steatosis, inflammation, or necrosis in any of the groups indicated no hepatotoxic effects from CPF or CPE supplementation. 

Histological analysis of the kidney sections in the control and CPF-supplemented mice (both male and female) showed normal renal corpuscles with glomeruli surrounded by narrow Bowman’s spaces and cortical tubules, indicating no adverse effects on renal structure. The tissue generally appears normal and healthy, with no significant signs of damage or disease. The glomeruli are well-formed, and the tubules are intact, with cells arranged in an orderly fashion. There is no evidence of inflammation or scarring. However, in CPE-supplemented mice, particularly females, there were subtle areas of pink luminal material accumulation within the tubules. These accumulations, potentially proteinaceous in nature, did not appear to disrupt the overall renal architecture, as the basic structure and organization of the kidney tissue remained intact. Histological analysis of the intestinal sections in the control and CPF-supplemented mice (both male and female) revealed no appreciable alterations in microvilli, suggesting that CPF does not negatively impact intestinal health. However, in CPE-supplemented mice (both male and female), the intestinal microvilli appear shortened and have thickened apices. These alterations in the intestinal morphology, while suggesting a potential adverse effect of CPE exposure on intestinal function, were not severe enough to significantly compromise nutrient absorption or barrier integrity. Interestingly, a current study demonstrated the acute non-toxicity of an ethanolic extract of coffee pulp in female mice, and histopathological evaluation showed no signs of toxicity or damage to the organs of the mice [73]. Moreover, recent research has reported no toxicity effects or significant changes in histological parameters of vital organs in rats after a single oral acute administration (2000 mg kg^−1^ bw) of coffee husk, parchment, and silverskin [12].

Based on the acute toxicity results, a 90-day sub-chronic toxicity study was conducted in female mice. No lethality was observed following oral administration of CPF or CPE. No visible changes were observed in skin, fur, eyes, or mucous membranes. After oral administration of CPF or CPE, increased agitation was observed in the mice, especially in the case of CPE. This behavior may be due to the high concentrations of caffeine in the extract compared to the flour. The final weight of mice in all groups (control, CPF, and CPE) was similar, with no significant differences observed (Figure 3A). This indicates that CPF and CPE supplementation did not significantly alter the overall body weight of the mice compared to the control group. The total weight gain at the end of the study was +4.4 g in the control group, +4.6 g in the CPF group, and +4.4 g in the CPE group (Figure 3B). There were no statistically significant differences in weight gain between the control and supplemented groups. However, analysis of the weight gain over the 90-day study period (Figure 3C) revealed that the CPE group experienced a slightly faster weight gain rate than the control and CPF groups, particularly during the initial weeks of the study. Despite this initial trend, this effect did not persist throughout the entire study period, resulting in similar final weights and total weight gains across all groups. On day 90, no significant differences were detected in the relative weights of control female mice’s heart, thymus, spleen, and kidneys compared to mice supplemented with CPF or CPE. However, the liver weights of supplemented mice were significantly lower than non-supplemented mice (Figure 3D). Change in the relative weight of organs is a reliable indicator that can be used in toxicological investigations to assess toxicity caused by drugs, food, or chemicals [72], although toxic changes should be validated through histopathological and biochemical studies. Hence, liver, kidney, and intestine histopathological analyses were conducted (Figure 3E). The liver histology of both CPF and CPE-treated mice appeared normal, with well-preserved hepatic architecture. Hepatic lobules were clearly defined, with hepatocytes arranged in cords radiating from central veins. Hepatocytes maintained their characteristic polygonal shape and contained centrally located, round-to-oval nuclei. The cytoplasm of the hepatocytes stained eosinophilic. There were no overt signs of steatosis (fatty change), inflammation, necrosis, or fibrosis in the liver tissue of either group. To further confirm that liver function was unaffected, biochemical parameters such as ALP, ALT, AST, GGT, and total bilirubin levels were measured, showing no significant differences between groups, indicating that the CPF and CPE diets did not alter liver function. 

The kidneys of CPF- and CPE-treated mice were generally histologically normal, with occasional multifocal alterations. Well-defined Bowman’s spaces surrounded glomeruli. While most renal corpuscles and tubules appeared normal, some variations in tubular appearance and glomerular structure were observed, particularly in the CPE group. In some CPE-treated mice, the eosinophilic material within the tubular lumen appears to form structures resembling hyaline casts. Additionally, subtle differences in tubular cellularity and occasional glomeruli that appeared slightly smaller or less defined were noted. However, these changes were not widespread or severe enough to disrupt the overall renal architecture. Renal function tests, including albumin, total protein, creatinine, blood urea nitrogen, inorganic phosphorus, and calcium levels, remained within normal ranges in all three groups. There were no statistically significant differences between the treatment groups (CPF and CPE) and the control group, suggesting that coffee pulp supplementation did not affect renal function under these conditions. In CPF-treated mice, the intestinal histology appears normal, with no evident abnormalities in the villi, epithelium, or lamina propria. In CPE-treated mice, the intestinal histology is generally normal, but subtle alterations are observed, including a slightly scalloped appearance of some villi tips and increased eosinophilic staining within the epithelial cells. Although suggestive of potential changes, these subtle alterations in intestinal structure observed in the CPE-treated mice do not appear severe enough to cause significant functional impairment based on this histological analysis alone. Additionally, analysis of lipid and hormonal profiles revealed interesting metabolic effects of CPF and CPE supplementation. As shown in Figure 3F, there were no significant differences in total cholesterol levels between the three groups. However, CPF and CPE diets significantly reduced LDL cholesterol levels compared to the control group (Figure 3G). This reduction in LDL cholesterol, a known risk factor for cardiovascular disease, could be attributed to the hypocholesterolemic effects of coffee pulp’s dietary fiber and antioxidant compounds, able to bind to cholesterol and bile salts in the gut (preventing lipids absorption and promoting its excretion) and to attenuate cholesterol synthesis and lipid accumulatio in hepatocytes [74], and may also be associated with the observed decrease in liver weight in CPF and CPE-treated mice. Notably, CPF and CPE diets significantly increased HDL cholesterol levels compared to the control group (Figure 3H), suggesting a potential additional benefit of these diets in promoting a more favorable lipid profile. No significant differences between the groups were observed in triglyceride levels (Figure 3I). The hormonal profile also showed significant changes with CPF and CPE supplementation. Although no significant differences were found in insulin levels, a notable increasing trend was observed from the control group to the CPF and CPE groups (Figure 3J). This trend aligns with previous studies that have explored the potential effects of coffee pulp and its bioactive compounds on insulin secretion and glucose metabolism since bioactive compounds in the coffee pulp may stimulate pancreatic insulin secretion [75] and increase glucose uptake and glucokinase activity in liver cells, potentially enhancing glucose sensitivity and utilization [11]. CPF and CPE diets significantly reduced leptin levels compared to the control group (Figure 3K). This reduction in leptin, a hormone associated with satiety and energy expenditure, may be linked to decreased liver weight and lower LDL cholesterol levels. Leptin, an adipokine primarily produced in adipose tissue, regulates food intake and energy expenditure. Leptin regulates lipid metabolism in the liver, increasing lipogenesis and decreasing fatty acid oxidation, thus leading to lower LDL cholesterol levels and a decrease in liver weight due to reduced fat accumulation [76]. The observed decrease in leptin levels suggests that CPF and CPE may promote satiety and enhance energy expenditure, potentially contributing to weight management and metabolic health.

Therefore, the toxicological study revealed that a single dose of sub-chronic 1000 mg kg^−1^ CPF and CPE for 90 days and a single acute dose of 2000 mg kg^−1^ of CPF and CPE after 14 days did not produce any significant changes in body weight nor any noteworthy alteration in the relative organ weight. Histopathological examination of vital organs from treated and control groups and microscopic examination of tissue sections did not reveal any pathology that could be attributed to CPF and CPE intake. Biochemical parameters of liver and kidney function remained within normal ranges, further supporting the safety of CPF and CPE consumption. Moreover, CPF and CPE diets induced meaningful changes in lipid and hormonal profiles, suggesting a potential positive impact on metabolic health. The toxicity of coffee pulp has not been investigated. However, the potential toxic effects of certain compounds widely present in coffee pulp, such as caffeoylquinic acids, have been evaluated. The consumption of caffeoylquinic acids from coffee by-products has not evidenced significant signs of toxicity or adverse effects, does not seem to pose a risk to human health, and can be considered safe after acute and sub-chronic oral exposure [60]. Additionally, CoffeeBerry^®^, a commercial product made from the whole coffee fruit, including coffee pulp, showed no adverse effects even at high doses after acute and sub-chronic administration in rats [77]. Despite no toxicological studies on coffee pulp, this material has traditionally been used in animal feed [78]. While the use of coffee pulp in animal feed has been controversial due to the presence of compounds like caffeine, chlorogenic, caffeic and tannic acids, tannins, and potassium, which may exhibit anti-physiological activity [79], our acute and sub-chronic toxicity evaluations provide valuable evidence supporting the safety of coffee pulp consumption. The absence of significant toxicological effects on organ weights, tissue histology, and biochemical parameters, coupled with the observed positive effects on lipid and hormonal profiles, strengthens the potential of coffee pulp as a safe and valuable resource, representing a promising first step in its valorization strategy and supporting its potential as valuable ingredient in human diets.

Whereas previous studies, including evaluations by the European Food Safety Authority (EFSA), have focused on the safety of coffee cherry pulp (cascara) primarily as an infusion ingredient [13,80], the toxicological effects of the CPF and the CPE for broader food applications have remained unexamined. This study fills that gap by conducting comprehensive acute and sub-chronic toxicity evaluations, providing new insights into the safety of these coffee pulp-derived ingredients. The high dietary fiber content, particularly the insoluble fiber in the CPF and soluble fiber in the CPE, makes both ingredients suitable candidates for incorporation into fiber-enriched food products. The CPF, with nearly 50% dietary fiber content, could serve as a sustainable alternative to traditional ingredients such as wheat flour or bran in bakery products, enhancing their nutritional profile [33]. For instance, the CPF’s fiber and protein content can improve the texture and dietary benefits of baked goods like bread or cookies, contributing to healthier, high-fiber alternatives [81]. The CPE’s significant concentration of bioactive compounds, including antioxidants like caffeoylquinic acids and caffeine, also highlights its potential in functional ready-to-use drinks. It could be utilized to formulate beverages that provide a rich source of antioxidants and a natural caffeine boost, positioning it as a healthier alternative to conventional caffeinated drinks [82]. Furthermore, given the amino acid profile and the protein quality observed in the CPF, it could be explored as a protein supplement in various food applications, especially in products designed to support muscle metabolism and overall nutritional intake [35,83]. These applications highlight the versatility and potential of coffee pulp derivatives for creating sustainable, health-promoting foods. Future studies could optimize product formulations, enhance bioavailability, and assess consumer acceptance of these coffee-pulp-enriched foods.

## 4. Conclusions

This study investigated the nutritional and phytochemical components present in the coffee pulp flour (CPF) and its aqueous extract (CPE), along with evaluating their in vivo toxicity in mice after acute and sub-chronic ingestion. The two ingredients were identified as a source of dietary fiber. The CPF mainly contained insoluble fiber, while the CPE consisted exclusively of soluble pectic polysaccharides. The total amino acid content and protein quality were higher in CPF, whereas the CPE exhibited higher concentrations of free amino acids, which could facilitate its bioavailability. Both matrices derived from coffee pulp showed elevated mineral content, while heavy-metal concentrations remained within acceptable limits. The CPF and the CPE exhibited a high concentration of caffeine and phenolic compounds, with a notable presence of gallic, protocatechuic, and 4-caffeoylquinic acids. Regarding acute and sub-chronic toxicity, the CPF and the CPE exhibited safety when orally administered to male and female mice. Both acute and sub-chronic doses were non-toxic, as they did not induce lethality or adverse effects in the mice, nor did they produce significant histopathological or biochemical adverse changes. Hence, this critical assessment brings about the potential of the CPF and the CPE as safe novel food ingredients with health-promoting properties. However, further studies are needed to explore their beneficial potential and understand their long-term effects.

## Figures and Tables

**Figure 1 foods-13-03006-f001:**
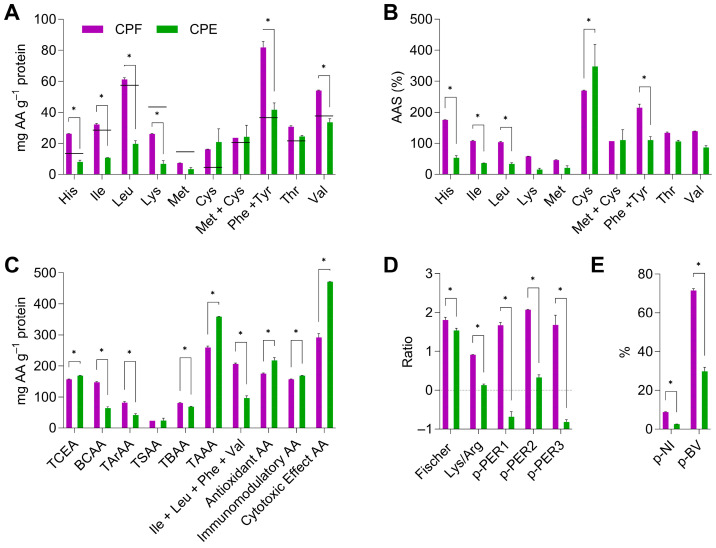
Assessment of protein quality in coffee pulp flour (CPF) and coffee pulp extract (CPE) based on essential amino acid (EAA) composition and derived parameters, including the concentrations of individual EAAs (mg AA g−1 protein) (**A**), where black horizontal lines indicate the amino acid scoring pattern (mg AA g^−1^ protein) of the reference protein (WHO/FAO/UNU Expert Consultation, 2007), the amino acid scores (AAS), expressed as the percentage of each EAA in the sample relative to its content in the reference protein (**B**), the concentration of total EAA (TEAA), total conditionally EAA (TCEA), total aromatic AA (TAraA), total sulfur AA (TSAA), total branched-chain AA (BCAA), and specific grouped AA and (mg AA g^−1^ protein) (**C**), the Fischer, Lys/Arg, predicted protein efficiency (p-PER) ratios (**D**), the predicted nutritional index (p-NI) and biological value (p-BV) for CPF and CPE (**E**). Data are presented as means ± SD. Asterisks (*) indicate significant differences (*p* < 0.05) between CPF and CPE according to *T*-test.

**Figure 2 foods-13-03006-f002:**
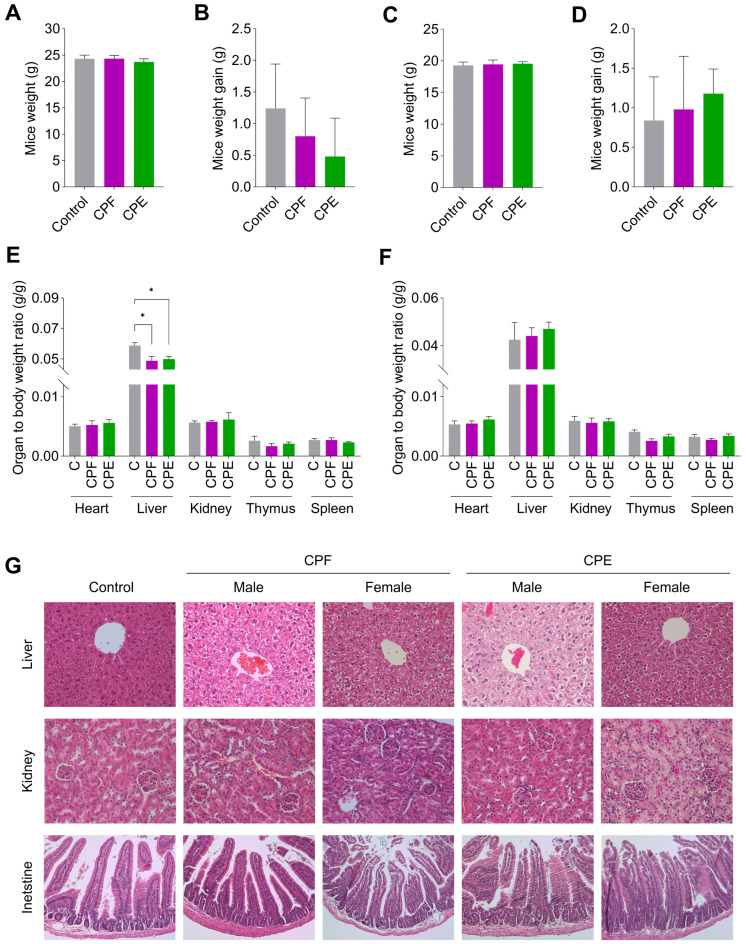
Effects of acute oral administration of coffee pulp flour (CSF) or coffee pulp extract CPE) on body weight, organ weight, and histology in mice, including body weight and weight gain in male (**A**,**B**) and female (**C**,**D**) mice after a 14-day acute toxicity test with 2000 mg kg^−1^ CPF or CPE. Relative organ weights of the heart, liver, kidney, thymus, and spleen of male (**E**) and female (**F**) mice collected 14 days after the acute toxicity test with a dose of 2000 mg kg^−1^. Representative histological images of liver, kidney, and intestine from control and treated (CPF or CPE) male and female mice (**G**). Data are presented as means ± SD. Asterisks (*) indicate significant differences (*p* < 0.05) compared to the control group when subject to the Dunnet test.

**Figure 3 foods-13-03006-f003:**
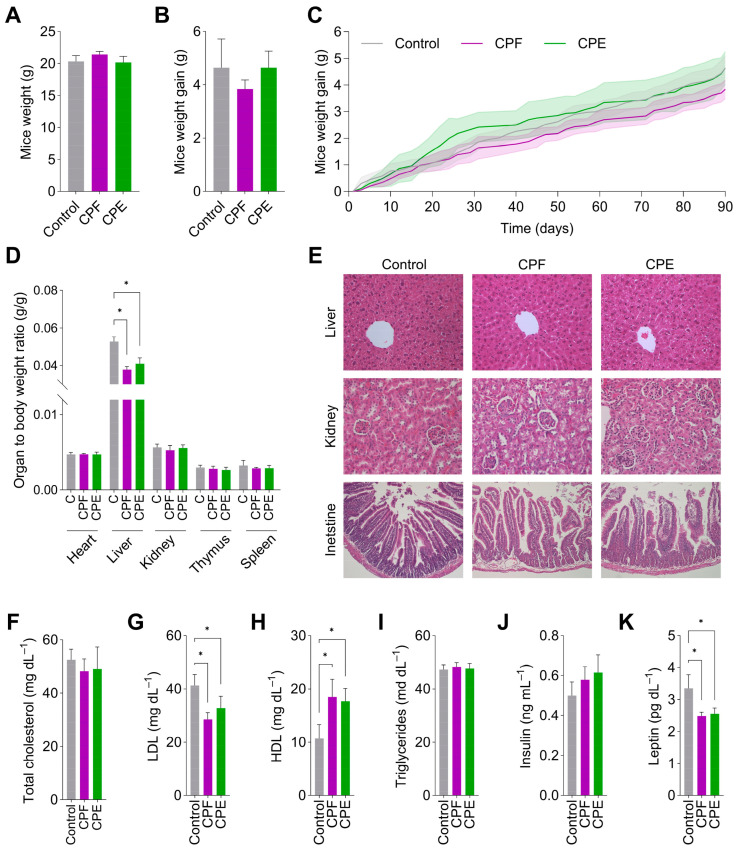
Effects of sub-chronic oral administration of coffee pulp flour (CPF) or coffee pulp extract (CPE) on body weight, organ weight, histology, and biochemical parameters in female mice, including body weight (**A**), weight gain (**B**), and in the evolution of body weight (**C**) in female mice during the 90-day sub-chronic toxicity test with a dose of 1000 mg kg^−1^. Relative organ weights in female mice, including heart, liver, kidney, thymus, and spleen, obtained after the sub-chronic toxicity test with a dose of 1000 mg/kg (**D**). Representative histological images of liver, kidney, and intestine from control and treated (CPF or CPE) female mice (**E**). Serum levels of total cholesterol (**F**), LDL cholesterol (**G**), HDL cholesterol (**H**), triglycerides (**I**), insulin (**J**), and leptin (**K**) after the sub-chronic toxicity test. Data are presented as means ± SD. Asterisks (*) indicate significant differences (*p* < 0.05) compared to the control group when subject to the Dunnet test.

**Table 1 foods-13-03006-t001:** Nutritional and chemical composition (g 100 g^−1^) of coffee pulp flour (CPF) and extract (CPE).

	CPF	CPE
**Total Carbohydrates**	76.7 ± 2.0	70.9 ± 2.9
Available Carbohydrates	26.9 ± 3.7 ^*^	42.9 ± 4.8
**Free sugars**	10.8 ± 1.3 ^*^	19.9 ± 1.2
Rhamnose	0.1 ± 0.0 ^*^	0.0 ± 0.0
Arabinose	1.6 ± 0.1 ^*^	3.0 ± 0.3
Xylose	0.0 ± 0.0 ^*^	0.2 ± 0.0
Fructose	7.6 ± 0.3 ^*^	14.0 ± 0.8
Glucose	1.6 ± 0.8 ^*^	2.8 ± 0.1
**Total Dietary Fiber**	49.8 ± 1.7 ^*^	28.0 ± 1.9
Soluble Dietary Fiber	11.2 ± 1.2 ^*^	28.0 ± 1.9
Insoluble Dietary Fiber	38.6 ± 0.5	n.d.
**Polysaccharides**	25.8 ± 1.8 ^*^	12.3 ± 1.1
Arabinose	3.6 ± 0.8 ^*^	1.4 ± 0.7
Xylose	1.6 ± 0.7	n.d.
Mannose	1.0 ± 0.1	n.d.
Galactose	1.6 ± 0.7	n.d.
Glucose	7.7 ± 0.5 ^*^	0.3 ± 0.0
Uronic acids	10.3 ± 0.9	10.7 ± 0.7
**Proteins**	12.9 ± 0.0 ^*^	7.5 ± 0.1
Total N	2.1 ± 0.0 ^*^	1.2 ± 0.0
**Fat**	2.8 ± 0.2 ^*^	3.8 ± 0.3
**Ash**	7.6 ± 0.1 ^*^	14.6 ± 0.0
**Energy (kJ)**	1188.3 ± 837.0	1274.4 ± 119.2
**Energy (Kcal)**	284.0 ± 20.0	304.6 ± 28.5

Results are reported as mean ± SD (*n* = 3). Mean values within rows followed by superscript asterisks (^*^) significantly differ (CPF vs. CPE) when subjected to *T*-test (*p* < 0.05). n.d.: Not detectable under the sensitivity of the methodology and equipment used.

**Table 2 foods-13-03006-t002:** Total and free amino acids content in coffee pulp flour (CPF) and extract (CPE) expressed as mg amino acids 100 g^−1^ of material (dw) and shorted by essential (EAAs) and non-essential (NEAAs) amino acids.

Amino Acids	Total Amino Acids		Free Amino Acids	
	CPF	CPE	CPF	CPE
Essential amino acids (EAA)				
Histidine	55.7 ± 1.7 ^hi*#^	9.8 ± 0.8 ^hi^	0.03 ± <0.1 ^d*^	0.08 ± <0.01 ^j#^
Isoleucine	68.3 ± 3.7 ^h*#^	13.3 ± 0.5 ^hi^	0.3 ± <0.1 ^d*^	1.0 ± <0.1 ^ghij#^
Leucine	130.0 ± 6.9 ^f*#^	24.1 ± 1.3 ^ghi^	0.2 ± <0.1 ^d*^	0.7 ± <0.01 ^ij#^
Lysine	55.5 ± 1.4 ^hi*#^	8.3 ± 2.8 ^hi^	0.01 ± <0.01 ^d*^	0.2 ± <0.1 ^ij#^
Methionine	15.4 ± 0.8 ^k*#^	4.1 ± 1.5 ^i^	0.04 ± <0.1 ^d^	0.03 ± <0.01 ^j#^
Phenylalanine	125.3 ± 5.5 ^fg*#^	40.2 ± 1.0 ^fg^	6.6 ± 0.8 ^b*^	2.4 ± 0.1 ^fgh#^
Threonine	65.2 ± 3.6 ^h*#^	29.9 ± 0.3 ^fgh^	0.1 ± <0.1 ^d*^	0.8 ± <0.1 ^ij#^
Valine	115.0 ± 4.7 ^g*#^	41.2 ± 0.9 ^fg^	0.9 ± <0.1 ^cd*^	2.5 ± <0.1 ^fg#^
Non-essential amino acids (NEAA)				
Alanine	203.4 ± 8.8 ^cd*#^	116.7 ± 9.8 ^d^	6.2 ± 0.7 ^b*^	14.9 ± 0.5 ^d#^
Arginine	60.7 ± 1.4 ^hi#^	65.8 ± 4.4 ^e^	2.0 ± 0.4 ^c*^	10.7 ± <0.1 ^e#^
Aspartic acid	333.2 ± 2.3 ^a#^	308.9 ± 14.8 ^a^	6.3 ± 1.0 ^b*^	30.3 ± 0.1 ^b#^
Cysteine	34.4 ± 1.0 ^j#^	25.4 ± 9.3 ^ghi^	0.4 ± <0.1 ^d*^	0.8 ± 0.1 ^hij#^
Glutamic acid	216.8 ± 11.5 ^bc*#^	131.7 ± 7.6 ^c^	0.6 ± 0.1 ^cd*^	3.2 ± 0.1 ^f#^
Glycine	224.3 ± 15.2 ^b*#^	51.4 ± 0.4 ^ef^	0.2 ± 0.1 ^d*^	1.8 ± <0.1 ^fghi#^
Proline	197.1 ± 4.2 ^d*#^	235.4 ± 21.6 ^b^	26.4 ± 2.2 ^a*^	74.2 ± 2.9 ^a#^
Serine	175.1 ± 6.8 ^e*#^	112.3 ± 8.4 ^d^	6.7 ± 1.4 ^b*^	27.1 ± <0.1 ^c#^
Tyrosine	48.4 ± 9.3 ^ij*#^	10.9 ± 2.0 ^hi^	0.1 ± <0.1 ^d*^	0.9 ± <0.1 ^ghij#^
∑AA	2123.7 ± 75.8 ^*#^	1229.3 ± 56.8	57.3 ± 6.8 ^*^	171.7 ± 3.5 ^#^
∑EAA	630.4 ± 28.3 ^*#^	170.8 ± 1.0	8.4 ± 0.8	7.7 ± <0.1 ^#^
∑NEAA	1493.3 ± 47.5 ^*#^	1058.6 ± 55.8	48.9 ± 6.0 ^*^	163.9 ± 3.5 ^#^
% EAA	29.7 ± 0.3 ^*#^	13.9 ± 0.6	14.6 ± 0.4 ^*^	4.5 ± 0.1 ^#^
% NEAA	70.3 ± 0.3 ^*#^	86.1 ± 0.6	85.4 ± 0.4 ^*^	95.5 ± 0.1 ^#^
EAA/NEAA	0.4 ± <0.1 ^*#^	0.2 ± <0.1	0.2 ± <0.1 ^*^	0.1 ± <0.1 ^#^
EAAI (%)	76.4 ± 0.8 ^*^	38.0 ± 2.0	-	-

Results are reported as mean ± SD (*n* = 3). Different superscript letters indicate significant differences within columns (Tukey’s test, *p* < 0.05). Superscript asterisks (*) and hash (#) symbols denote significant differences (*T*-test, *p* < 0.05) in TAA (either FAA) content between or within matrices, respectively.

**Table 3 foods-13-03006-t003:** Mineral composition of coffee pulp flour (CPF) and extract (CPE) grouped by macroelements, microelements, and contaminants.

Element	CPF	CPE
Macroelements (mg g^−1^ sample, dw)		
Calcium (Ca)	2.4 ± 0.2 ^b*^	5.1 ± 0.2 ^b^
Magnesium (Mg)	0.9 ± 0.1 ^b*^	1.5 ± <0.1 ^c^
Phosphorus (P)	1.2 ± 0.1 ^b*^	0.6 ± <0.1 ^c^
Potasium (K)	24.2 ± 1.7 ^a*^	35.1 ± 1.3 ^a^
Microelements (µg g^−1^ sample, dw)		
Boron (B)	22.7 ± 1.7 ^b*^	6.0 ± 4.2 ^b^
Copper (Cu)	17.6 ± 1.2 ^b*^	0.2 ± 0.2 ^b^
Iron (Fe)	76.9 ± 1.0 ^b*^	19.5 ± 2.9 ^b^
Manganese (Mn)	62.0 ± 6.0 ^b*^	15.3 ± 0.5 ^b^
Silicon (Si)	504.4 ± 136.7 ^a*^	102.0 ± 22.9 ^a^
Zinc (Zn)	17.3 ± 0.5 ^b^	18.6 ± 18.1 ^b^
Contaminants (µg 100 g^−1^ sample, dw)		
Arsenic (As)	18.2 ± <0.1 ^b*^	n.d.
Cadmiun (Cd)	2.5 ± 2.3 ^b^	6.1 ± 6.1 ^b^
Chromium (Cr)	136.6 ± <0.1 ^a^	90.2 ± 33.2 ^a^
Lead (Pb)	n.d.	0.5 ± 0.5 ^b^
Nickel (Ni)	105.4 ± 7.3 ^a^	53.0 ± 33.7 ^ab^

Results are reported as mean ± SD (*n* = 3). The different superscript letter indicates significant differences between the element content within their respective element group by matrix (Tukey’s test, *p* < 0.05). Superscript asterisks denote significant differences in the corresponding mineral content between matrices following *T*-test analysis (*p* < 0.05). n.d.: Not detectable under the sensitivity of the methodology and equipment used.

**Table 4 foods-13-03006-t004:** Concentration of individual phenolic compounds and caffeine (mg 100 g^−1^) in coffee pulp flour (CPF) and extract (CPE).

Compounds	CPF	CPE
**Phenolic compounds**		
Hydroxybenzoic acids		
3,4,5-Trihydroxybenzoic acid (Gallic acid)	46.9 ± 2.0 ^c*^	68.5 ± 4.0 ^c^
3,4-Dihydroxybenzoic acid (Protocatechuic acid)	175.7 ± 0.7 ^a*^	312.2 ± 7.5 ^a^
Hydroxycinnamic acids		
3-(3′,4′-Dihydroxycinnamoyl)quinic acid (3-Caffeoylquinic acid, 3-CQA)	5.3 ± 0.6 ^ef*^	12.1 ± 1.0 ^d^
4-(3′,4′-Dihydroxycinnamoyl)quinic acid (4-Caffeoylquinic acid, 4-CQA) (*cis*)	6.6 ± 0.5 ^de*^	12.7 ± 0.6 ^d^
4-(3′,4′-Dihydroxycinnamoyl)quinic acid (4-Caffeoylquinic acid, 4-CQA) (*trans*)	85.3 ± 1.4 ^b*^	145.0 ± 0.4 ^b^
5-(3′,4′-Dihydroxycinnamoyl)quinic acid (5-Caffeoylquinic acid, 5-CQA)	3.9 ± 0.3 ^fgh^	n.d.
5-(4′-Hydroxy-3′-methoxycinnamoyl)quinic acid (5-Feruloylquinic acid, 5-FQA)	2.0 ± 0.1 ^h*^	8.7 ± 0.1 ^d^
3,5-bis(3′,4′-Dihydroxycinnamoyl)quinic acid (3,5-Dicaffeoylquinic acid, 3,5-diCQA)	4.9 ± 0.1 ^efg*^	7.0 ± 0.1 ^d^
5-(4′-Hydroxycinnamoyl)quinic acid (5-*p*-Coumaroylquinic acid, 5-CoQA)	2.8 ± 0.0 ^gh^	n.d.
Flavones		
5,7,4′-Trihydroxyflavone-6,8-di-C-glucoside (Apigenin-6,8-di-C-glucoside)	6.4 ± 0.2 ^de*^	8.0 ± 1.1 ^d^
Flavonols		
3,3′,4′,5,7-Pentahydroxyflavone 3,7-di-β-glucoside (Quercetin-3,7-di-*O*-glucoside)	2.6 ± 0.1 ^h^	n.d.
3,3′,4′,5,7-Pentahydroxyflavone-3-rutinoside (Quercetin-3-*O*-rutinoside)	7.9 ± 0.2 ^d*^	14.5 ± 0.2 ^d^
3,3′,4′,5,7-Pentahydroxyflavone 3-β-glucoside (Quercetin-3-*O*-glucoside)	6.1 ± 0.3 ^def*^	8.9 ± 0.2 ^d^
Total phenolic compounds	356.4 ± 6.5 ^*^	597.6 ± 15.2
**Methylxanthines**		
1,3,7-Trimethylxanthine (Caffeine)	473.1 ± 6.1 ^*^	787.9 ± 34.3

Results are reported as mean ± SD (*n* = 3). Mean values within columns followed by different superscript letters (a, b, c, d, e, f, g, h) are significantly different when subjected to Tukey’s test (*p* < 0.05). Mean values within rows followed by superscript asterisks (*) significantly differ (CPF vs. CPE) when subjected to *T*-test (*p* < 0.05). n.d.: Not detectable under the sensitivity of the methodology and equipment used.

## Data Availability

The original contributions presented in the study are included in the article, further inquiries can be directed to the corresponding author.

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
