# Peer review of "Unveiling the Nutritional Profile and Safety of Coffee Pulp as a First Step in Its Valorization Strategy"

_foods, 2024, doi:10.3390/foods13183006_

Round 1
Reviewer 1 Report
Comments and Suggestions for Authors
The given paper lacks novelty as numerous studies already examine the chemical composition of coffee pulp, including its amino acid profile and nutritional value. Existing literature extensively covers the nutritional and bioactive components of coffee pulp, making the current study's contribution marginal. Additionally, EFSA's evaluation included reviewing toxicological studies to identify any potential adverse health effects. The results showed no significant safety concerns related to the consumption of dried coffee pulp under the specified conditions. Hence, this aspect is not novel too. Therefore, the findings presented do not offer significant new insights to the field.
Also, the discussion can be improved. More papers can be cited and related to this research.
No comments
Author Response
- The given paper lacks novelty as numerous studies already examine the chemical composition of coffee pulp, including its amino acid profile and nutritional value. Existing literature extensively covers the nutritional and bioactive components of coffee pulp, making the current study’s contribution marginal. Additionally, EFSA’s evaluation included reviewing toxicological studies to identify any potential adverse health effects. The results showed no significant safety concerns related to the consumption of dried coffee pulp under the specified conditions. Hence, this aspect is not novel too. Therefore, the findings presented do not offer significant new insights to the field.
- Author’s response: Thank you for your comment. While we acknowledge that some studies have explored the general chemical composition of coffee pulp, this research addresses key gaps in the literature and presents several novel contributions.
Firstly, no previous studies have focused on the coffee pulp’s detailed amino acid profile. Our study is the first to comprehensively analyze the amino acid composition, which is essential for understanding its full nutritional value. Furthermore, we analyzed two distinct forms of coffee pulp: flour (CPF) and an aqueous extract (CPE). The study of the extract represents a new area of investigation, which continues the authors’ field of study, as it has not been the focus of previous research.
In terms of safety, the European Food Safety Authority (EFSA) has conducted evaluations related to the use of coffee cherry pulp (cascara) as an ingredient in beverages, these assessments were limited to its use in infusions and did not encompass the broader toxicological evaluation of coffee pulp when used as a food ingredient. Our study goes beyond this by providing comprehensive acute and sub-chronic toxicity assessments specifically for the CPF and the CPE. These detailed toxicological studies offer critical safety data necessary for evaluating the potential of coffee pulp as a functional food ingredient. This focus on both the flour and extract forms provides new insights into their safety and potential applications, which were not covered in previous EFSA evaluations or existing literature.
To further highlight the novelty of this research, we have included a new paragraph in the introduction that explicitly addresses these contributions.
- Paragraph modified/added (page 3, lines 89-103): While the potential of the coffee pulp as a bioactive ingredient has been recognized, specific aspects, such as its amino acid and mineral profiles and toxicological safety, have not been previously examined in two key ingredients from the coffee pulp: flour (CPF) and extract (CPE). Therefore, this study aimed to fill those gaps in the knowledge of coffee pulp by validating its nutritional value and assessing its safety for consumption, thereby establishing a foundation for its potential valorization as a bioactive food ingredient. The comprehensive characterization of its chemical composition and acute and sub-chronic toxicity assays will provide a more holistic understanding of the coffee pulp’s potential for application in the food industry.
- Also, the discussion can be improved. More papers can be cited and related to this research.
-
- Author’s response: Thank you for your feedback. In response to your suggestion, we have enhanced the discussion by incorporating additional relevant citations that link our findings to existing literature. These references further contextualize the novelty and significance of our work in relation to previous studies. While the European Food Safety Authority (EFSA) has evaluated the safety of coffee cherry pulp (cascara) as an ingredient primarily for infusions, our study provides the first comprehensive acute and sub-chronic toxicity evaluations of coffee pulp flour (CPF) and extract (CPE) for broader food applications. We have cited recent studies on the nutritional benefits of fiber and bioactive compounds found in coffee pulp and their potential for use in bakery and beverage products. These additions help to position coffee pulp as a valuable and versatile ingredient for functional foods, building on both our findings and the work already published in the field.
- Paragraph modified/added (page 25 and 26, lines 990-1004): Whereas previous studies, including evaluations by the European Food Safety Authority (EFSA), have focused on the safety of coffee cherry pulp (cascara) primarily as an infusion ingredient [13,80], the toxicological effects of the CPF and the CPE for broader food applications have remained unexamined. This study fills that gap by conducting comprehensive acute and sub-chronic toxicity evaluations, providing new insights into the safety of these coffee pulp-derived ingredients. The high dietary fiber content, particularly the insoluble fiber in the CPF and soluble fiber in the CPE, makes both ingredients suitable candidates for incorporation into fiber-enriched food products. The CPF, with nearly 50% dietary fiber content, could serve as a sustainable alternative to traditional ingredients such as wheat flour or bran in bakery products, enhancing their nutritional profile [33]. For instance, the CPF’s fiber and protein content can improve the texture and dietary benefits of baked goods like bread or cookies, contributing to healthier, high-fiber alternatives [81]. The CPE’s significant concentration of bioactive compounds, including antioxidants like caffeoylquinic acids and caffeine, also highlights its potential in functional ready-to-use drinks. It could be utilized to formulate beverages that provide a rich source of antioxidants and a natural caffeine boost, positioning it as a healthier alternative to conventional caffeinated drinks [82]. Furthermore, given the amino acid profile and the protein quality observed in the CPF, it could be explored as a protein supplement in various food applications, especially in products designed to support muscle metabolism and overall nutritional intake [35,83]. These applications highlight the versatility and potential of coffee pulp derivatives for creating sustainable, health-promoting foods. Future studies could optimize product formulations, enhance bioavailability, and assess consumer acceptance of these coffee-pulp-enriched foods.
-
- References added:
European Food Safety Authority Technical Report on the Notification of Cherry Pulp from Coffea Arabica L. and Coffea Canephora Pierre Ex A. Froehner as a Traditional Food from a Third Country Following Article 14 of Regulation (EU) 2015/2283. EFSA Support. Publ. 2021, 18, doi:10.2903/sp.efsa.2021.EN-6657.
Turck, D.; Bohn, T.; Castenmiller, J.; De Henauw, S.; Hirsch-Ernst, K.I.; Maciuk, A.; Mangelsdorf, I.; McArdle, H.J.; Naska, A.; Pelaez, C.; et al. Safety of dried coffee husk (cascara) from Coffea arabica L. as a Novel food pursuant to Regulation (EU) 2015/2283. EFSA J. 2022, 20, e07085, doi:10.2903/J.EFSA.2022.7085.
Rosas-Sánchez, G.A.; Hernández-Estrada, Z.J.; Suárez-Quiroz, M.L.; González-Ríos, O.; Rayas-Duarte, P. Coffee Cherry Pulp By-Product as a Potential Fiber Source for Bread Production: A Fundamental and Empirical Rheological Approach. Foods 2021, 10, 742, doi:10.3390/foods10040742.
Rivas-Vela, C.I.; Amaya-Llano, S.L.; Castaño-Tostado, E. Effect of Extrusion Process on the Obtention of a Flour from Coffee Pulp Coffea Arabica Variety Red Caturra and Its Use in Bakery Products. J. Food Sci. Technol. 2023, 60, 2792–2801, doi:10.1007/s13197-023-05797-x.
Sánchez-Martín, V.; López-Parra, M.B.; Iriondo-DeHond, A.; Haza, A.I.; Morales, P.; del Castillo, M.D. Instant Cascara: A Potential Sustainable Promoter of Gastrointestinal Health. In Proceedings of the ICC 2023; MDPI: Basel Switzerland, 2023; p. 21.
Patil, S.; Pimpley, V.; Warudkar, K.; Murthy, P.S. Valorisation of Coffee Pulp for Development of Innovative Probiotic Beverage Using Kefir: Physicochemical, Antioxidant, Sensory Analysis and Shelf Life Studies. Waste and Biomass Valorization 2022, 13, 905–916, doi:10.1007/s12649-021-01554-3.
Iriondo-Dehond, M.; Iriondo-Dehond, A.; Herrera, T.; Fernández-Fernández, A.M.; Sorzano, C.O.S.; Miguel, E.; Del Castillo, M.D. Sensory Acceptance, Appetite Control and Gastrointestinal Tolerance of Yogurts Containing Coffee-Cascara Extract and Inulin. Nutrients 2020, 12, doi:10.3390/nu12030627.
-
Reviewer 2 Report
Comments and Suggestions for Authors
This article is very complete in terms of scientific content. The authors studied coffee pulp and its aqueous extract regarding physicochemical, nutritional and toxicological parameters. The writing is focused, with replicable methodological details and a logical sequence in the results presented. The results are robust and detailed, with appropriate discussions and clarifications of the points found. The toxicological part worked with a relatively small number of mice, 5 in each treatment and control with males and females, which is a worldwide trend in animal research. However, the results obtained for acute and subchronic toxicity are relevant, not only for the macroscopic aspects, but also for the cytological aspects presented. Therefore, the article is ready for publication.
Author Response
- This article is very complete in terms of scientific content. The authors studied coffee pulp and its aqueous extract regarding physicochemical, nutritional and toxicological parameters. The writing is focused, with replicable methodological details and a logical sequence in the results presented. The results are robust and detailed, with appropriate discussions and clarifications of the points found. The toxicological part worked with a relatively small number of mice, 5 in each treatment and control with males and females, which is a worldwide trend in animal research. However, the results obtained for acute and subchronic toxicity are relevant, not only for the macroscopic aspects, but also for the cytological aspects presented. Therefore, the article is ready for publication.
- Author’s response: Thank you for your positive feedback and support. We are pleased that you found the study comprehensive and the results relevant. Your comments are greatly appreciated.
Reviewer 3 Report
Comments and Suggestions for Authors
Dear authors, the paper is well-written and seems valuable for the enhancement of coffee pulp as a byproduct of the coffee industry. I only have a few suggestions/comments to make below.
Line 144 - Was the HPLC-PAD method employed to analyze neutral sugars based on any published article? Is it an internal method? What are the main performance figures if it is internal? The same question applies to the HPLC-DAD-ESI/MSn method for the analysis of phenolic compounds and methylxanthines (line 211).
Regarding abbreviations like CPE and CPF, please write them out in full the first time in the text with the abbreviation in parentheses, and then use only the abbreviation afterward. Ensure uniformity throughout the manuscript, and apply this rule to all abbreviations. What does the abbreviation SDF stand for?
In subchapter 3.4, "CPE Exhibited a Higher Concentration of (Poly)phenols and Caffeine Than CPF," starting at line 611, you could compare the results with other food matrices to better elucidate the readers.
Lastly, I suggest paying closer attention to the uniformity of the references.
Author Response
- Line 144 - Was the HPLC-PAD method employed to analyze neutral sugars based on any published article? Is it an internal method? What are the main performance figures if it is internal? The same question applies to the HPLC-DAD-ESI/MSn method for the analysis of phenolic compounds and methylxanthines (line 211).
- Author’s response: Thank you for your comment. Both the HPLC-PAD method for neutral sugar analysis and the HPLC-DAD-ESI/MSn method for phenolic compounds and methylxanthines were based on published methodologies, with no internal adaptations. We have now cited the appropriate references in the manuscript to provide clarity.
- Paragraph modified/added (page 4, lines 136-144): HPLC-PAD was employed to analyze the neutral sugar composition in CPF and CPE. Hydrolysates were neutralized using AG4-X4 resin, and sugars were analyzed using a microguard column (Aminex Carbo-P) in series with a carbohydrate analysis column (Aminex HPX-87P). Sugars were quantified using standard sugars, with erythritol as the internal standard [20]. The concentration of uronic acids was determined using a commercial kit (K-URONIC, Megazyme Co. Wicklow, Ireland).
- Paragraph modified/added (page 7, lines 262-264): For (poly)phenols and methylxanthines analysis, a Hewlett-Packard-1100 HPLC chromatograph equipped with a diode array detector (DAD) and a quaternary pump, made by Agilent Technologies in Palo Alto, CA, USA, was utilized [9].
- Regarding abbreviations like CPE and CPF, please write them out in full the first time in the text with the abbreviation in parentheses, and then use only the abbreviation afterward. Ensure uniformity throughout the manuscript, and apply this rule to all abbreviations. What does the abbreviation SDF stand for?
- Author’s response: Thank you for your suggestion. We have double-checked the manuscript to ensure that all abbreviations, including SDF (soluble dietary fiber), IDF (insoluble dietary fiber), and TDF (total dietary fiber), are properly defined upon first mention. CPF (coffee pulp flour) and CPE (coffee pulp extract) are also defined at their first appearance and used consistently throughout the manuscript for clarity and conciseness.
- Paragraph modified/added (page 3, lines 89-95): While the potential of the coffee pulp as a bioactive ingredient has been recognized, specific aspects, such as its amino acid and mineral profiles and toxicological safety, have not been previously examined in two key ingredients from the coffee pulp: flour (CPF) and ex-tract (CPE). Therefore, this study aimed to fill those gaps in the knowledge of coffee pulp by validating its nutritional value and assessing its safety for consumption, thereby establishing a foundation for its potential valorization as a bioactive food ingredient.
- Paragraph modified/added (page 4, lines 128-133): Total dietary fiber (TDF), the sum of insoluble dietary fiber (IDF) and soluble dietary fiber (SDF) was determined by the enzymatic-gravimetric method (Mes-Tris AOAC method 991.43) with slight modifications.
- In subchapter 3.4, "CPE Exhibited a Higher Concentration of (Poly)phenols and Caffeine Than CPF," starting at line 611, you could compare the results with other food matrices to better elucidate the readers.
- Author’s response: Thank you for your valuable suggestion. While polyphenols are present in a wide variety of plant-based foods, for clarity and relevance, we have limited the comparison to other coffee by-products, which share a more similar composition and processing context. This narrower focus allows for a more accurate comparison of phenolic and caffeine concentrations, given that coffee by-products, such as silverskin, parchment, husk, and spent coffee grounds, undergo similar environmental and processing factors.
- Paragraph modified/added (page 20, lines 773-792): Overall, the diversity and distribution of bioactive compounds in coffee pulp are consistent with findings in other coffee by-products, such as coffee silverskin, parchment, husk, and spent coffee grounds. Studies have highlighted caffeine and chlorogenic acids as predominant components in these by-products. For instance, coffee silverskin and husk aqueous extracts contain caffeine levels of 1922 and 982 mg/100 g and chlorogenic acid concentrations of 279 and 346 mg/100 g, respectively [68]. Acetone-water extracts from spent coffee grounds report caffeine and chlorogenic acid levels of 83 and 147 mg/100 g [69], which are comparable to the concentrations observed in our study. Additionally, coffee husk demonstrated a rich phenolic profile, including significant amounts of protocatechuic and gallic acids [68]. In contrast, coffee parchment exhibited a lower chlorogenic acid content (29 mg/100 g) but contained other notable phenolic compounds, such as vanillic and p-coumaric acids [70]. These results highlight the consistent presence of phenolic compounds and caffeine across coffee by-products, reinforcing the potential of coffee pulp, particularly CPF and CPE, as valuable ingredients for functional foods and nutraceuticals. The higher phenolic and caffeine concentrations in CPE suggest its suitability for developing value-added products, especially for their antioxidant and stimulant properties.
- References added:
Rebollo-Hernanz, M.; Zhang, Q.; Aguilera, Y.; Martín-Cabrejas, M.A.; Gonzalez de Mejia, E. Relationship of the Phytochemicals from Coffee and Cocoa By-Products with their Potential to Modulate Biomarkers of Metabolic Syndrome In Vitro. Antioxidants 2019, 8, 279. https://doi.org/10.3390/antiox8080279
Bouhzam, I.; Cantero, R.; Margallo, M.; Aldaco, R.; Bala, A.; Fullana-i-Palmer, P.; Puig, R. Extraction of Bioactive Compounds from Spent Coffee Grounds Using Ethanol and Acetone Aqueous Solutions. Foods 2023, 12, 4400. https://doi.org/10.3390/foods12244400
Aguilera, Y.; Rebollo-Hernanz, M.; Cañas, S.; Taladrid, D.; Martín-Cabrejas, M.A. Response Surface Methodology to Optimise the Heat-Assisted Aqueous Extraction of Phenolic Compounds from Coffee Parchment and Their Comprehensive Analysis. Food & Function 2019, 10, 3967-3977. https://doi.org/10.1039/C9FO00544G
- Lastly, I suggest paying closer attention to the uniformity of the references.
- Author’s response: Thank you for your observation. We have thoroughly reviewed and revised the references to ensure uniformity and consistency in formatting according to the journal's guidelines.
Reviewer 4 Report
Comments and Suggestions for Authors
line 85 information about species and variety(s) coffee providing the coffee pulp should be included.
Discussion later in the manuscript mentions the variability that can be expected based on soil, environmental and genetic variability so further description of the site should be included.
How was the coffee pulp collected handled after collection in Nicaragua.
How many pulp samples were collected. Are they the basis for replication for the analysis or were all replications produced from one pulp sample.
line 86 was the pulp dry or wet before ball grinding into flour?
An impressive study of elemental and biochemical analysis of coffee pulp fractions followed by two feeding studies with results expressed in terms of human intake then compared with nutritional and toxicological guidelines.
Author Response
- Line 85 information about species and variety(s) coffee providing the coffee pulp should be included.
- Author’s response: Thank you for your observation. We have now included the specific species and variety of the coffee used in this study.
- Paragraph modified/added (page 3, lines 106-114): The coffee pulp used in this study was sourced from “Las Morenitas” farm, located in the highlands of Nicaragua, northwest of Jinotega, at an altitude of 1200 meters (13.2082 −85.8871). The pulp was mechanically separated from the cherries of the Arabica species, the variety Caturra, through the wet processing method. After collection, the raw and sun-dried coffee pulp was packaged in sealed bags and shipped at room temperature. The coffee pulp was ground into flour in a pilot-scale ball mill (Ortoalresa-Álvarez Redondo S.A., Madrid, Spain) for 72 h to produce the CPF and stored in sealed containers at -20 °C until needed.
- Discussion later in the manuscript mentions the variability that can be expected based on soil, environmental and genetic variability so further description of the site should be included.
- Author’s response: We have addressed the need for more detailed information about the site, describing the farm, its location, and the environmental factors that may contribute to variability in the coffee pulp. This information has been incorporated into the Discussion section, expanding on the environmental factors that may influence the chemical composition of coffee pulp.
- Paragraph modified/added (page 18, lines 702-717): This variance in the concentration of chlorogenic acid and other bioactive compounds may be influenced by several factors, including coffee species or variety, the degree of ripeness, local climate, soil characteristics, agricultural practices, and post-harvest processing methods [60]. The coffee pulp used in this study was sourced from the “Las Morenitas” farm in the highlands of Nicaragua, situated at an altitude of 1200 meters with a tropical humid climate and temperatures ranging from 15 to 26°C year-round. These environmental factors and sustainable agricultural practices significantly impact the coffee pulp quality and biochemical composition. Soil conditions, such as nutrient availability, organic matter, and pH, along with the surrounding vegetation, such as cedar and guava trees, contribute to the microenvironment that affects the chemical composition of the coffee pulp, influencing its flavor and nutritional profile through subtle metabolic interactions [61,62].
- References added:
Mengesha, D.; Retta, N.; Woldemariam, H.W.; Getachew, P. Changes in Biochemical Composition of Ethiopian Coffee Arabica with Growing Region and Traditional Roasting. Front. Nutr. 2024, 11, doi:10.3389/fnut.2024.1390515.
Ahmed, S.; Brinkley, S.; Smith, E.; Sela, A.; Theisen, M.; Thibodeau, C.; Warne, T.; Anderson, E.; Van Dusen, N.; Giuliano, P.; et al. Climate Change and Coffee Quality: Systematic Review on the Effects of Environmental and Management Variation on Secondary Metabolites and Sensory Attributes of Coffea Arabica and Coffea Canephora. Front. Plant Sci. 2021, 12, doi:10.3389/fpls.2021.708013.
- How was the coffee pulp collected handled after collection in Nicaragua.
- Author’s response: We appreciate this comment and have clarified the handling of the coffee pulp after its collection.
- Paragraph modified/added (page 3, lines 109-111): After collection, the raw and sun-dried coffee pulp was packaged in sealed bags and shipped at room temperature.
- How many pulp samples were collected. Are they the basis for replication for the analysis or were all replications produced from one pulp sample.
- Author’s response: Thank you for pointing this out. We have clarified the number of samples used and the replication process.
- Paragraph modified/added (page 3, lines 119-121): From a single batch of coffee pulp (25 kg), 5 kg were milled to produce the CPF, which was subsequently used to prepare the CPE. All replications for the analysis were performed using this single batch of coffee pulp.
- Line 86 was the pulp dry or wet before ball grinding into flour?
- Author’s response: We have now clarified the condition of the coffee pulp prior to grinding. The coffee pulp sourced from wet processing was sun-dried before being ground into flour using a ball mill.
- Paragraph modified/added (page 3, lines 109-111): After collection, the raw and sun-dried coffee pulp was packaged in sealed bags and shipped at room temperature.
- An impressive study of elemental and biochemical analysis of coffee pulp fractions followed by two feeding studies with results expressed in terms of human intake then compared with nutritional and toxicological guidelines.
- Author’s response: Thank you for your positive feedback. The comparison of the results with nutritional and toxicological guidelines was a key aspect of this study, ensuring that the findings are not only scientifically sound but also relevant for human consumption. We believe this comprehensive approach provides valuable insights into the potential for safely incorporating coffee pulp into food products.
Reviewer 5 Report
Comments and Suggestions for Authors
Dear authors,
Please consider the following points to correct the manuscript:
L37-39 - Specify at which part of coffee cherry processing the pulp is generated. Include information on how much of pulp is generated in comparison to the beans processed.
L72-82 - There are plenty of research on the valorization of coffee waste products. Please mention specifically the novelties of this research.
L114-116 - Include reference of methodology used. Please double check the need of including the place where analysis was done.
L127-139 - Include reference of methodology used.
L172 - It is confusing. Was the HCl solution consisted of 50% or 37% HCl?
Topic 2.6.1 - Include a table with the diets for each group (control, CPE and CPF).
Table1 -Insoluble dietary fiber/CPE - Is the ''-'' the same as non detectable (nd)?
L327-328 I recommend refer the ''nd' meaning: Not detectable under the methodology and equipment used.
L329 - What is SDF? Soluble dietary fiber? If yes, explain the acronym in text.
l351 - What is the range mentioned in literature? Include.
L810-811- In this case, should not make any difference in comparison with the control? Please be more specific in the text.
L862 - Is that ''CoffeeBerry®' a commercial product or are you referring to the coffee cherries?
Author Response
- L37-39 - Specify at which part of coffee cherry processing the pulp is generated. Include information on how much of pulp is generated in comparison to the beans processed.
- Author’s response: Thank you for your comment. We have clarified that coffee pulp is generated during the wet processing of coffee cherries, where the outermost layer (the pulp and skin) is mechanically removed after washing. Coffee pulp constitutes approximately 41% of the cherry’s weight, meaning about 1 ton of pulp is produced for every 2 tons of green coffee beans. This makes coffee pulp a significant by-product of coffee production, which has traditionally been regarded as waste.
- Paragraph modified/added (page 2, lines 41-48): The coffee pulp, comprising the outermost layer of the coffee cherry and representing approximately 41% of the cherry’s weight, is produced during the wet processing when the pulp and skin are removed after washing [3]. The coffee pulp has traditionally been regarded as a waste product, with around 1 ton generated for every 2 tons of green coffee beans produced [4].
-
- References added:
Klingel, T.; Kremer, J.I.; Gottstein, V.; Rajcic de Rezende, T.; Schwarz, S.; Lachenmeier, D.W. A Review of Coffee By-Products Including Leaf, Flower, Cherry, Husk, Silver Skin, and Spent Grounds as Novel Foods within the European Union. Foods 2020, 9, 665, doi:10.3390/foods9050665.
- L72-82 - There are plenty of research on the valorization of coffee waste products. Please mention specifically the novelties of this research.
- Author’s response: Thank you for your comment. We acknowledge the substantial body of research on coffee waste products; however, this study offers several novel contributions. First, it provides the first detailed analysis of coffee pulp’s amino acid and mineral profiles, specifically examining both the flour (CPF) and extract (CPE) forms, which have not been studied previously in this context. Additionally, while previous research has primarily focused on the safety of coffee cherry products, our study conducts comprehensive acute and sub-chronic toxicity assessments specifically on coffee pulp, offering more direct insights into its safety for human consumption. Furthermore, including the coffee pulp extract (CPE) adds a unique perspective to the valorization process, as most studies focus on the general pulp without examining extract-specific applications. These aspects collectively highlight the novelty of this research and its contribution to the field of coffee waste valorization.
- Paragraph modified/added (page 3, lines 89-103): While the potential of the coffee pulp as a bioactive ingredient has been recognized, specific aspects, such as its amino acid and mineral profiles and toxicological safety, have not been previously examined in two key ingredients from the coffee pulp: flour (CPF) and extract (CPE). Therefore, this study aimed to fill those gaps in the knowledge of coffee pulp by validating its nutritional value and assessing its safety for consumption, thereby establishing a foundation for its potential valorization as a bioactive food ingredient. The comprehensive characterization of its chemical composition and acute and sub-chronic toxicity assays will provide a more holistic understanding of the coffee pulp’s potential for application in the food industry.
- L114-116 - Include reference of methodology used. Please double check the need of including the place where analysis was done. L127-139 - Include reference of methodology used.
- Author’s response: We have now provided the necessary references for the methodologies employed, particularly for the amino acid extraction and analysis.
- Paragraph modified/added (page 4, lines 147-159): For the extraction of total amino acids, CPF (~40 mg) and CPE (~15 mg) were mixed with 200 µL of 6 mol L−1 HCl and subjected to 110 °C for 21 h to achieve acid hydrolysis of the proteins [21]. Afterward, the samples were vacuum-dried, weighed, and stored at –20 °C for further analysis. Norleucine (6 nmol for CPF and 20 nmol for CPE) was added as an internal standard before hydrolysis to normalize amino acid recovery.
-
- Paragraph modified/added (page 4, lines 160-162): The extraction of free amino acids was performed as previously reported with minor adjustments [22]. Briefly, 150 mg of CPF or CPE were frozen with N2 in a mortar.
- Paragraph modified/added (page 5, lines 175-178): Thus, the samples (extracts of total and free amino acids) were examined by ion exchange chromatography and post-column online derivatization with ninhydrin using a Biochrom 30+ amino acid analyzer following manufacturer instructions (Biochrom, Massachusetts, EEUU).
-
- References added:
Machado, M.; Machado, S.; Pimentel, F.B.; Freitas, V.; Alves, R.C.; Oliveira, M.B.P.P. Amino Acid Profile and Protein Quality Assessment of Macroalgae Produced in an Integrated Multi-Trophic Aquaculture System. Foods 2020, 9, 1382, doi:10.3390/foods9101382.
Hacham, Y.; Avraham, T.; Amir, R. The N-Terminal Region of Arabidopsis Cystathionine γ-Synthase Plays an Important Regulatory Role in Methionine Metabolism. Plant Physiol. 2002, 128, 454–462, doi:10.1104/pp.010819.
- L172 - It is confusing. Was the HCl solution consisted of 50% or 37% HCl?
- Author’s response: Thank you for pointing this out. To clarify, the HCl solution used in the study consisted of 37% HCl diluted (50:50, v/v) with water. We have revised the manuscript to ensure this is clearly stated.
- Paragraph modified/added (page 6, lines 218-223): Afterward, ashes were carefully mixed with ultrapure water (2.5 mL) and an 18.5% HCl solution, covered with a glass watch to avoid material losses, and submitted to 80 °C for 30 min.
- Topic 2.6.1 - Include a table with the diets for each group (control, CPE and CPF).
- Author’s response: Thank you for your comment. We have now included the necessary details about the diet and the administration of CPF and CPE. Mice were fed ad libitum with a standard diet containing 51.7% carbohydrates, 21.4% protein, 5.1% lipids, 3.9% fiber, 5.7% minerals, and 12.2% humidity (SafeA03; Safe Augy, France). The diet was the same for all three groups, with only the differing supplementation: the control group received vehicle gelatin cubes, while the CPF and CPE groups were administered gelatin cubes containing CPF or CPE. Ingestion of the gelatin cubes was voluntary after a training period where the animals were conditioned to accept them, ensuring accurate dosing during the toxicity assays.
- Paragraph modified/added (page 7 and 8, lines 298-304): Mice were fed ad libitum with a standard diet containing 51.7% carbohydrates, 21.4% protein, 5.1% lipids, 3.9% fiber, 5.7% minerals, and 12.2% humidity (SafeA03; Safe Augy, France). The diet was the same for all groups, with only the supplementation differing: the control group received plain gelatin cubes, while the CPF and CPE groups were administered gelatin cubes containing CPF or CPE. Fresh drinking water was available ad libitum.
- Table1 -Insoluble dietary fiber/CPE - Is the “-” the same as non detectable (nd)?
- Author’s response: Thank you for your comment. We confirm that the “-” in the table represents “non-detectable (nd)” under the methodology used.
- Paragraph modified/added (page 9 and 10, line 384):
Table 1. Nutritional and chemical composition (g 100 g−1) of coffee pulp flour (CPF) and extract (CPE).
|
|
CPF |
CPE |
|
Total Carbohydrates |
76.7 ± 2.0 |
70.9 ± 2.9 |
|
Available Carbohydrates |
26.9 ± 3.7 |
42.9 ± 4.8 |
|
Free sugars |
10.8 ± 1.3* |
19.9 ± 1.2 |
|
Rhamnose |
0.1 ± 0.0* |
0.0 ± 0.0 |
|
Arabinose |
1.6 ± 0.1* |
3.0 ± 0.3 |
|
Xylose |
0.0 ± 0.0* |
0.2 ± 0.0 |
|
Fructose |
7.6 ± 0.3* |
14.0 ± 0.8 |
|
Glucose |
1.6 ± 0.8* |
2.8 ± 0.1 |
|
Total Dietary Fiber |
49.8 ± 1.7* |
28.0 ± 1.9 |
|
Soluble Dietary Fiber |
11.2 ± 1.2* |
28.0 ± 1.9 |
|
Insoluble Dietary Fiber |
38.6 ± 0.5 |
n.d. |
|
Polysaccharides |
25.8 ± 1.8* |
12.3 ± 1.1 |
|
Arabinose |
3.6 ± 0.8* |
1.4 ± 0.7 |
|
Xylose |
1.6 ± 0.7 |
n.d. |
|
Mannose |
1.0 ± 0.1 |
n.d. |
|
Galactose |
1.6 ± 0.7 |
n.d. |
|
Glucose |
7.7 ± 0.5* |
0.3 ± 0.0 |
|
Uronic acids |
10.3 ± 0.9 |
10.7 ± 0.7 |
|
Proteins |
12.9 ± 0.0* |
7.5 ± 0.1 |
|
Total N |
2.1 ± 0.0* |
1.2 ± 0.0 |
|
Fat |
2.8 ± 0.2* |
3.8 ± 0.3 |
|
Ash |
7.6 ± 0.1* |
14.6 ± 0.0 |
|
Energy (kJ) |
1188.3 ± 837 |
1274.4 ± 119.2 |
|
Energy (Kcal) |
284.0 ± 20.0 |
304.6 ± 28.5 |
Results are reported as mean ± SD (n = 3). Mean values within rows followed by superscript asterisks (*) significantly differ (CPF vs. CPE) when subjected to T-test (p < 0.05). n.d.: Not detectable under the sensitivity of the methodology and equipment used.
- L327-328 I recommend refer the “nd” meaning: Not detectable under the methodology and equipment used.
- Author’s response: Thank you for the suggestion. We have clarified the meaning of “n.d.” in all tables, updating it to “Not detectable under the sensitivity of the methodology and equipment used.” This ensures consistency and clarity throughout the manuscript, and we have revised the table legends accordingly to reflect this definition.
- Paragraph modified/added (page 10, lines 389-391; page 16, lines 600-601; page 19, lines 731-733): n.d.: Not detectable under the sensitivity of the methodology and equipment used.
- L329 - What is SDF? Soluble dietary fiber? If yes, explain the acronym in text.
- Author’s response: Thank you for pointing this out. SDF indeed refers to Soluble Dietary Fiber, and we have added an explanation for the acronym in the text.
- Paragraph modified/added (page 4, lines 128-133): Total dietary fiber (TDF), the sum of insoluble dietary fiber (IDF) and soluble dietary fiber (SDF) was determined by the enzymatic-gravimetric method (Mes-Tris AOAC method 991.43) with slight modifications [20].
- L351 - What is the range mentioned in literature? Include.
- Author’s response: We have now included the range mentioned in the literature to provide a clearer context for the discussion.
- Paragraph modified/added (page 10, lines 414-418): Concerning proteins, a significantly higher amount was recorded for CPF (12.9%) compared to CPE (7.5%), which is in agreement with the literature (7.9–10.7%, depending on the processing method) [34,35].
- L810-811- In this case, should not make any difference in comparison with the control? Please be more specific in the text.
- Author’s response: Thank you for the comment. We have revised the text to clarify that, under these specific conditions, the difference between the treatment and control groups is not statistically significant.
- Paragraph modified/added (page 24, lines 915-923): Renal function tests, including albumin, total protein, creatinine, blood urea nitrogen, inorganic phosphorus, and calcium levels, remained within normal ranges in all three groups. There were no statistically significant differences between the treatment groups (CPF and CPE) and the control group, suggesting that coffee pulp supplementation did not affect renal function under these conditions.
- L862 - Is that “CoffeeBerry®” a commercial product or are you referring to the coffee cherries?
- Author’s response: Thank you for your observation. “CoffeeBerry®” is indeed a commercial product. We have clarified this in the manuscript.
- Paragraph modified/added (page 25, lines 975-980): Besides, CoffeeBerry®, a commercial product made from the whole coffee fruit, including coffee pulp, showed no adverse effects even at high doses after acute and sub-chronic administration in rats.